# High-throughput telomere length measurement at nucleotide resolution using the PacBio high fidelity sequencing platform

Cheng-Yong Tham [1,17], LaiFong Poon [2,17], TingDong Yan [2,3], Javier Yu Peng Koh[2], Muhammad Khairul Ramlee[2], Vania Swee Imm Teoh [1], Suihan Zhang [4], Yi Cai[2,5], Zebin Hong[6], Gina S. Lee[7], Jin Liu [8,9], Hai Wei Song[6], William Ying Khee Hwang[2,10,11], Bin Tean Teh [2,6,12], Patrick Tan[1,2,13,14], Lifeng Xu [4], Angela S. Koh [7], Motomi Osato [1,15] ✉ & Shang Li [2,16] ✉

Telomeres are specialized nucleoprotein structures at the ends of linear chromosomes. The progressive shortening of steady-state telomere length in normal human somatic cells is a promising biomarker for age-associated diseases. However, there remain substantial challenges in quantifying telomere length due to the lack of high-throughput method with nucleotide resolution for individual telomere. Here, we describe a workflow to capture telomeres using newly designed telobaits in human culture cell lines as well as clinical patient samples and measure their length accurately at nucleotide resolution using single-molecule real-time (SMRT) sequencing. Our results also reveal the extreme heterogeneity of telomeric variant sequences (TVSs) that are dispersed throughout the telomere repeat region. The presence of TVSs disrupts the continuity of the canonical (5′-TTAGGG-3′)n telomere repeats, which affects the binding of shelterin complexes at the chromosomal ends and telomere protection. These findings may have profound implications in human aging and diseases.

The ends of human chromosomes are capped by telomeres that consist of long tracts of hexa-nucleotide DNA repeats (5′-TTAGGG-3′)n with short single-stranded G-rich 3′ overhangs that are protected by associated proteins, including the shelterin complex[1,2]. Telomeres are essential for the stability as well as the complete replication of the human genome[3,4]. Telomeres are synthesized by the enzyme telomerase, a reverse-transcriptase containing two core subunits—a protein subunit, hTert, and an RNA subunit, hTR/hTER[5,6]. While hTR/hTER is ubiquitously expressed, the expression of *hTERT* is repressed and eventually silenced during cellular differentiation, except in germ cells and stem cells. The lack of telomerase activity in somatic cells results in the progressive shortening of telomeres (by about 50–200 bp) during each cell division due to the incomplete replication of chromosomal ends by DNA polymerase and telomere end processing[7,8]. When telomeres are critically short, they become dysfunctional and induce DNA

damage responses leading to cellular senescence, which serves as an important barrier to tumorigenesis in long-lived mammals, including humans[9–12].

While pluripotent stem cells have robust telomerase activity that is sufficient to support their continuous proliferation[13–16], adult tissue stem/progenitor cells only express limited amount of *hTERT* necessary to support their self-renewal within the adult lifespan. The progressive shortening of telomeres in adult tissue stem/progenitor cells inevitably limits their proliferation capacity and contributes directly to human aging[17–19]. Previous studies have suggested that genetic factors play a major role in telomere length variations in the general population. Genetic mutations in telomere- and telomerase-associated genes are known to be associated with telomere diseases/telomere syndromes/telomeropathies[20,21], which are characterized by premature aging caused by accelerated telomere shortening. However, other non-

genetic factors, such as life stressors and lifestyles, can further regulate telomere length homeostasis by modulating telomere attrition rate during normal aging and diseases[22]. Recent advancements in the field have shown that greater overall telomere attrition predicts aging-related diseases and mortality in the general population as well[22]. These results highlight the importance of telomere homeostasis in human health and raise the prospect of using telomere length as a predictive biomarker for human aging and diseases. Telomere length in peripheral blood leukocytes (PBLs) or peripheral blood mono-nuclear cells (PBMCs) has frequently been used as a biomarker in aging-related research. It reflects not only the progressively shortening of telomeres and diminished self-renewal capability of hematopoietic stem and progenitor cells (HSPCs), but also telomere maintenance in other tissues[22]. Therefore, accurate measurement of telomere length in PBLs and PBMCs in the population not only provides a biomarker for aging, but also a necessary tool for developing potential interventions to slow down aging.

Several existing methods have been implemented in the laboratories and clinical settings for measuring telomere length. For instance, Terminal Restriction Fragment (TRF) analysis, which is the gold standard for telomere length measurement, involves the use of frequent-cutting restriction enzymes, such as RsaI, HinfI, MnlI, or HphI, that does not cut within the telomeric repeats (5′-TTAGGG-3′)n, followed by Southern hybridization[23]. Another method that is widely used in the clinical diagnosis of patients with telomere diseases is Flow-FISH[24]. This method uses fluorescence-labeled peptide nucleic acid (PNA) probes that specifically bind telomere repeats for fluorescence in situ hybridization (FISH), followed by fluorescence measurement using flow cytometry to measure mean telomere length in a single cell. Furthermore, real-time quantitative polymerase chain reaction (qPCR)-based methods enable for rapid and high-throughput telomere length measurement[25], and have become the method of choice in large epidemiological studies. These methods provide a readout of mean and relative telomere length for a pool of telomeres, but they lack resolution at the level of individual telomere.

As the shortest telomeres, rather than the average telomere length, predicts viability and long-term proliferation capacity of tissue stem cells[26], measuring the length of the shortest telomeres serve as a more relevant biomarker for aging-related studies. Therefore, several methods have been developed to provide relative quantification of telomere length at individual chromosomal ends including quantitative fluorescence in situ hybridization (Q-FISH) using telomere-specific PNA probes on metaphase cell preparation[27]; telomere length combing assay (TCA) to measure the length of telomeres on stretched chromosome fibers[28]; and PNA hybridization and analysis of single telomere (PHAST) that detects single telomeres passing through microfluidic channels[29]. In addition to fluorescence in-situ hybridization, techniques that can characterize the length distribution of individual telomere have also been developed. These include the single telomere length analysis (STELA) method and its derivatives U-STELA and TeSLA, which employ adaptor ligation and PCR amplification, followed by Southern hybridization. Using these methods, telomeres (along with some subtelomeric sequences) as short as 0.3 kb can be readily detected in human cancer cells as well as pre-senescent fibroblast cells. The presence of extremely short telomeres (<1000 bp) in pre-senescent primary fibroblast cells correlates well with SA-β-gal staining, a biochemical marker for cellular senescence. However, these procedures are laborious, technically demanding, and have low throughput. Recent developments of techniques like single telomere absolute-length rapid (STAR) assay using droplet digital PCR[30], high-throughput single telomere length analysis (HT-STELA)[31], and high-throughput quantitative fluorescence in situ hybridization (HT-Q-FISH)[32] provide further improvements for measuring individual telomere length at a large scale. Regardless, these methods can only provide the relative length but not the absolute length of an individual

telomere at nucleotide resolution. Most importantly, they lack the ability to detect telomeric variant sequences (TVSs) that may have profound implications in human aging and diseases.

Telomere attrition rate, which affects the steady-state telomere length in individual, is known to vary throughout the lifespan of humans. Previous studies have indicated that the age-related telomere shortening in PBMCs was estimated to be 1190 bp in early life, 126 bp in childhood and 43 bp in adulthood per year respectively[33]. Given the slow and progressive shortening of telomeres in adulthood, a highly sensitive method that can longitudinally detect telomere attrition in an individual in time is needed for both experimental and clinical studies[34]. In this regard, the recent development of long-read sequencing platforms has enabled the direct sequencing of telomere-containing genomic DNA fragments to obtain absolute telomere length at individual chromosome end[35–39]. However, current methods require the sequencing of whole genome to analyze the individual's telomere distribution, which precludes high-throughput capability. In addition, these methods failed to sustain the integrity of 3′ telomeric single-stranded G-rich overhangs for telomere length measurement during library processing.

To overcome these problems, we design biotinylated telobaits that are complementary to the telomeric single-stranded G-rich overhangs at the 3′ ends of chromosomes. Following ligation, we purify the intact, full-length telomere-containing genomic DNA fragments using streptavidin-conjugated magnetic beads and subject the purified DNA fragments to SMRT sequencing. Embedding unique barcodes within the telobaits allows multiplexing of samples and, in turn, permits high-throughput telomere length measurement at nucleotide resolution using the PacBio high fidelity (HiFi) sequencing platform. We demonstrate that this sequencing platform provides an ideal method for highly accurate absolute telomere length measurement of individual telomere in culture cell lines as well as clinical patient samples. In addition, our sequencing data also reveal the presence of telomeric variant sequences (TVSs) interspersed within the long tracts of canonical telomeric repeat regions. The distribution and sequence of TVSs are unique to each individual. Most importantly, the TVSs can affect the binding density of shelterin complexes in the telomeric region and in turn, affect telomere protection at chromosomal ends. Furthermore, the unique distribution of TVSs may act as an inherited non-coding polymorphism with implications in human aging and diseases.

## Results

### Accuracy of PacBio HiFi sequencing of DNA vector with telomeric repeats

The PacBio HiFi sequencing platform leverages on the optimization of circular consensus sequencing (CCS) to achieve >99.5% accuracy for single-molecule real-time (SMRT) sequencing of long reads averaging 10–25 kb[40,41]. Previous publications have provided a glimpse into telomere repeats at the chromosome ends, although with very low coverage depth[42–44]. They have reported that the telomeric repeat regions are interspersed with variant sequences exhibiting extreme heterogenicity, which may be related to the sequencing kinetics in repetitive DNA loci. To validate whether the PacBio HiFi sequencing platform can be used as a reliable tool for measuring telomere length, we first performed HiFi sequencing on a linearized DNA vector (pWY82) that contains direct repeats of the *Arabidopsis* telomeric sequence 5′-TTTAGGG-3′. As shown in supplementary Fig. S1a, we obtained highly accurate sequencing data of the plasmid. The average accuracy of the CCS reads for the DNA vector has a median of more than Q40 (99.99%) and a mean of Q29 (99.87%). Subsequently, only reads with at least Q20 were used for alignment analysis. The alignment accuracy for the non-repetitive region of the DNA vector was close to 100%, while the alignment accuracy for the repetitive region was >95% (Fig. S1b). Overall, the sequencing accuracy in the non-repetitive vector backbone is better than the repetitive telomeric sequence region. However,

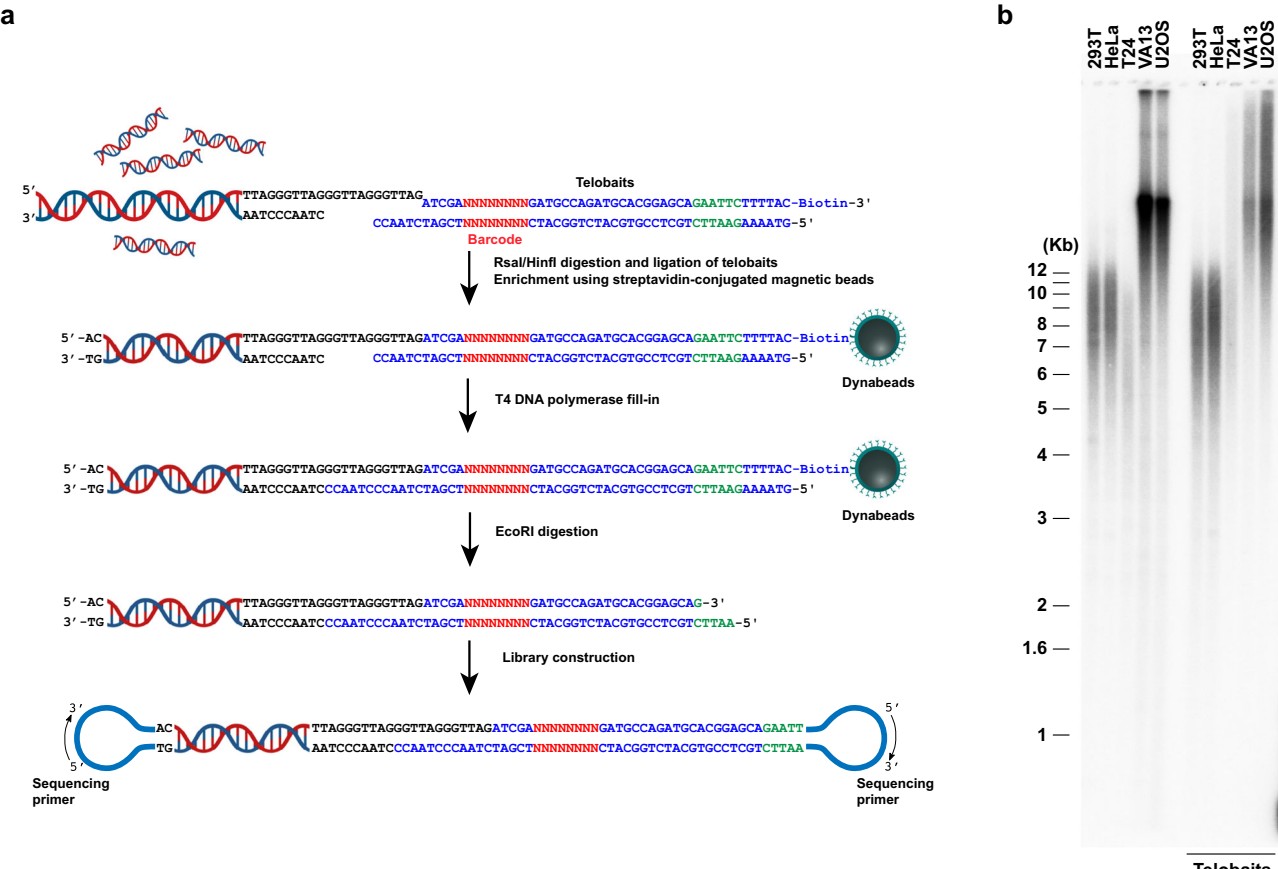

**Fig. 1 | Enrichment of telomere-containing genomic DNA fragments using telobaits for PacBio HiFi sequencing. a** Schematic illustration of the key steps in the enrichment of telomere repeat-containing genomic DNA fragments for PacBio HiFi sequencing. **b** Teloblot results showing that the telomere-containing genomic DNA fragments enriched following telobaits ligation (without trimming using RsaI and HinfI restriction endonucleases) and eluted after EcoRI digestion have similar length distributions as genomic DNA fragments digested directly using EcoRI restriction endonuclease. Representative teloblot results from one of the two independent experiments are shown (Source Data 1).

sequencing errors at the individual nucleotide level for the direct repeats of *Arabidopsis* telomeric sequence were similar to those found in the non-repetitive regions of the plasmid (Fig. S1c). Therefore, our results concur with previously published reports on genome assembly using PacBio HiFi sequencing, and further validate PacBio HiFi sequencing as a reliable method for SMRT sequencing of telomere-containing genomic DNA fragments.

## Telomere enrichment using telobaits

Telomeres contain ~5–15 kb of double-stranded (5′-TTAGGG-3′)n repeats that terminate with a single-stranded G-rich 3′ overhang, which is about 12–300 nucleotides in length[8,45]. To preserve full-length telomere-containing genomic DNA fragments for PacBio HiFi sequencing, we devised a method for telomere enrichment and purification. High-molecular-weight genomic DNA was extracted and digested with RsaI and HinfI restriction endonucleases, which do not cut within the repetitive telomeric 5′-TTAGGG-3′ repeats, to trim down the telomere-containing genomic DNA fragments for optimized PacBio HiFi sequencing (Fig. 1a). To enrich for the telomere-containing genomic DNA fragments, we adopted a similar strategy that was previously used to profile the telomere G-rich telomere overhangs[8,45] with several specific modifications. In total, we designed six telobaits each containing a single telomere repeat with all six possible ends that can anneal to the telomeric single-stranded 3′ overhangs (Fig. S2a). Each telobait also contains a unique EcoRI restriction enzyme site and 3′ biotin labeling. Complementation and

subsequent ligation of the telobaits to the G-tails preserve the full-length telomeres for telomere length measurement (Fig. S2a). Following ligation, T4 DNA polymerase was used to fill in the single-stranded gap between the telobait and the double-stranded telomere DNA (Fig. 1a). The telomere-containing genomic DNA fragments were subsequently purified using Dynabeads MyOne Streptavidin T1 magnetic beads and eluted following EcoRI restriction endonuclease digestion (Fig. 1a). As shown in Fig. 1b, the telomere-containing genomic DNA fragments that were enriched by telobaits ligation (without RsaI and HinfI restriction endonuclease digestion) and eluted via EcoRI digestion showed comparable length distributions to the genomic DNA digested directly using EcoRI (uncropped teloblot raw data can be found in source data 1). The recovery efficiency of telomere-containing genomic DNA fragments using this method is >20% (Fig. S2b, c), which is in line with the previous report[8]. The incorporation of barcodes in the telobaits further permits multiplexing of samples for high-throughput telomere length measurements (Figs. 1a and S2a, and Supplementary Data 1 and 2). Together, these results show the consistency of employing this method in enriching telomeres and accurately measuring their length using PacBio HiFi sequencing.

## Telomere length measurement with nucleotide resolution

Next, we use the devised method to enrich for telomere-containing genomic DNA fragments from culture cell lines for telomere length measurement using PacBio HiFi sequencing. We observed that a

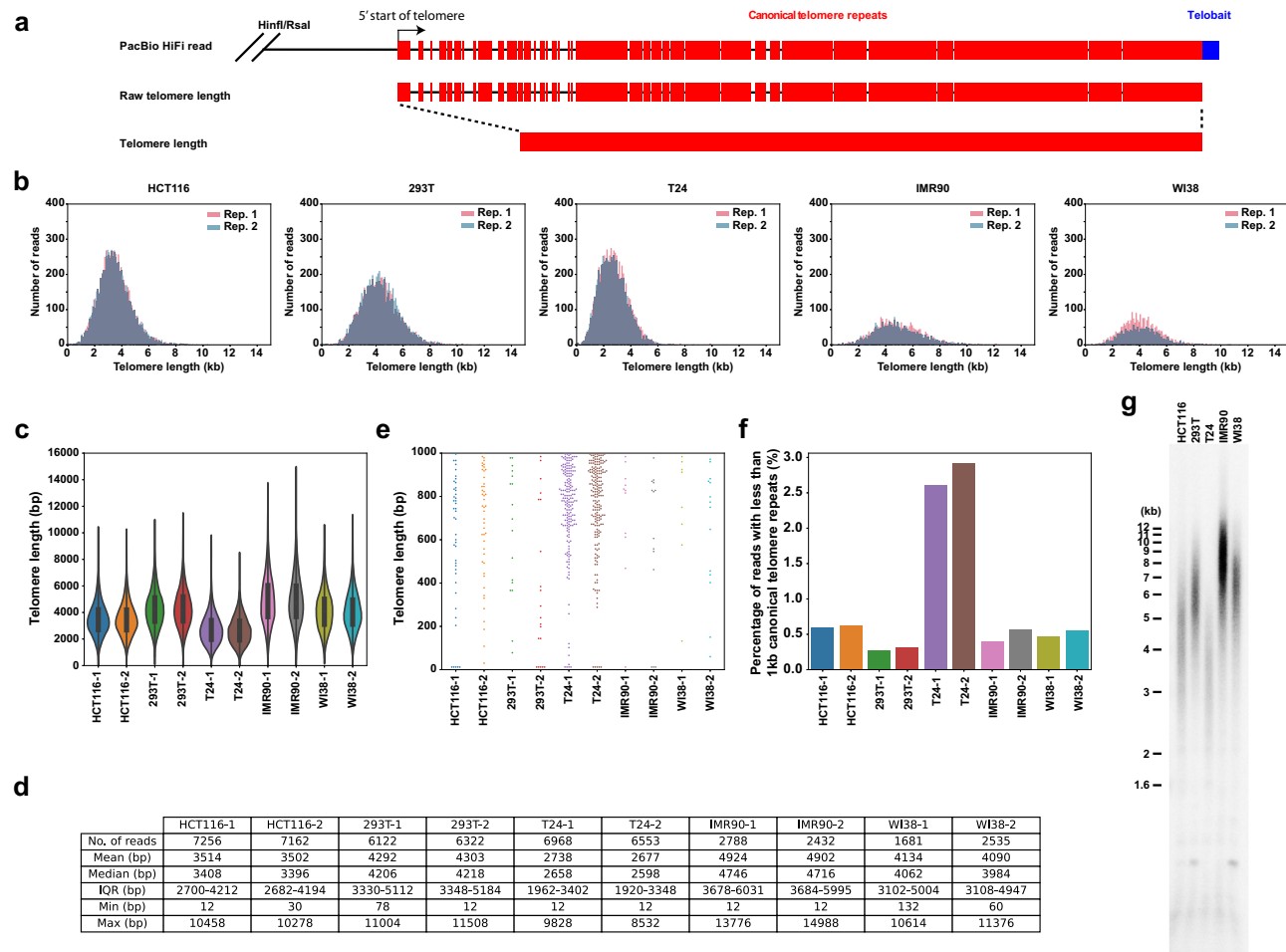

**Fig. 2 | Telomere length measurement in culture cells using PacBio HiFi sequencing. a** Schematic illustration for the estimation of raw telomere length or telomere length with or without the inclusion of heterogenous TVSs in sequencing reads, respectively. **b, c** Histograms and violin plots of telomere length distribution in HCT116, 293T, T24, IMR90 (PD47), and WI38 (PD40) cells obtained from two biological replicates (Rep.) in a single sequencing run. PD population doubling. Within each violin, the white center circle denotes the median value, the bounds of box represent the 25th to 75th percentile values, the whiskers represent adjacent values within 1.5 interquartile range, the ends of the whiskers depict the minimum and maximum values within the range, and the violin shape reflect the kernel density plot of the entire dataset. Source data are provided in Source Data 2. **d.** Table showing the characteristics of telomere length distribution in HCT116, 293T, T24, IMR90 (PD47) and WI38 (PD40) cells obtained from two biological replicates as shown in the violin plot in **c. e** Beeswarm plot showing the over-representation of extremely short telomeres (<1000 bp) in sequencing reads, specifically from T24 cells. Source data are provided in Source Data 2. **f** Bar graph showing the over-representation of extremely short telomeres (<1000 bp) in sequencing reads, specifically in T24 cells after normalization. **g** Teloblot of HCT116, 293T, T24, IMR90 (PD47), and WI38 (PD40) genomic DNA digested with the same HinfI and RsaI restriction endonucleases that were used to trim down the genomic DNA fragment during telomere enrichment for PacBio HiFi sequencing. Representative results from one of the two independent experiments are shown (Source Data 1).

typical PacBio HiFi sequencing read of telomere-containing genomic DNA contains TVSs with high heterogenicity within the telomere repeat regions (Fig. S3). Furthermore, our findings imply that the traditional TRF method for telomere length measurement could frequently overestimate the actual length of the telomere hexanucleotide repeats in a sample due to the presence of TVSs. To estimate the mean telomere length, we arbitrarily defined the 5′ start of telomere as a sequence that contains at least two consecutive telomeric repeats (5′TTAGGGTTAGGG-3′), given that the minimum binding motif for shelterin complex contains 1.5 canonical repeats (5′-TTAGGGTTA-3′)[46], and the 3′ end of a telomere to be the end of its 3′ overhang tail. The distance between these two markers is referred to as raw telomere length (Fig. 2a), which is comparable to the traditional definition of telomere length (from the start of TTAGGG repeat to the chromosome end). The telomere-associated shelterin complex that protects the chromosomal ends from being recognized as double-stranded DNA breaks, binds to canonical telomere repeats in a sequence-specific manner and hence is predicted to be disrupted by the presence of TVSs. Therefore, we also calculated the

length of telomere containing solely the canonical (5′-TTAGGG-3′)n repeats, which we defined as telomere length (Fig. 2a).

We quantified the telomere lengths of telomere-containing DNA fragments extracted from HCT116, 293T, T24, IMR90 (PD47), and WI38 (PD40) cells in independent biological replicates. The distribution of telomere lengths was very similar between biological replicates in a single sequencing run (Fig. 2b, c) or in separate sequencing runs (Fig. S4a, b). The estimated mean telomere length in replicates showed very high concordance with nucleotide resolution (Figs. 2d and S4c). Previous studies have shown that T24 cells contain extremely short telomeres[42]. Consistent with this, our analysis showed that T24 cells have a higher proportion of extremely short telomeres (<1000 bp) as compared to other cell lines (Fig. 2e, f). The mean telomere length in cultured cell lines as estimated using PacBio HiFi sequencing is consistent with the telomere length estimated using TRF (Fig. 2g; uncropped teloblot raw data can be found in source data 1). The shortest telomeres that we observed from PacBio HiFi sequencing contained only two telomere repeats (12nt). This is likely due to the artifact from RsaI digestion as

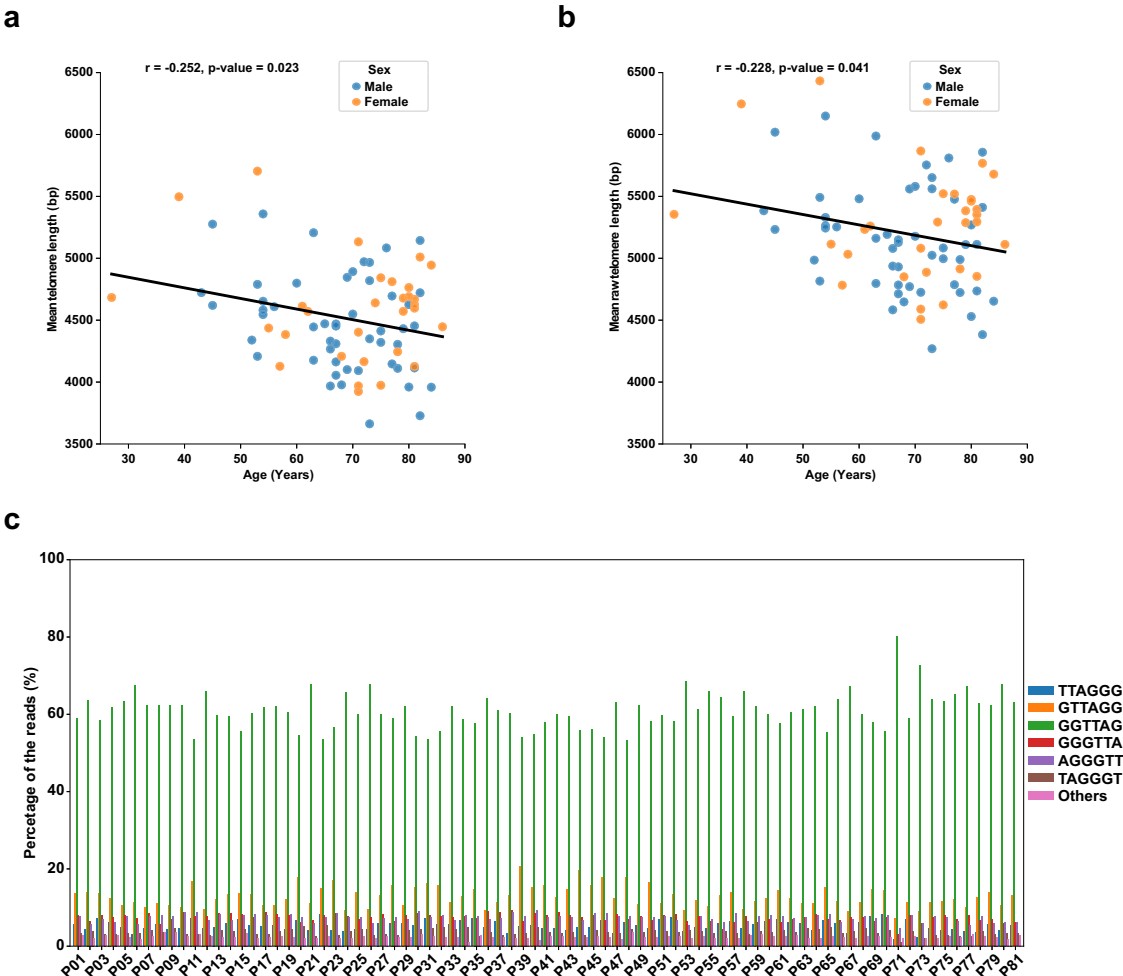

**Fig. 3 | High-throughput telomere length measurement in patient PBL samples.**
**a** Mean telomere length distribution of patient PBL samples shows age-dependent shortening as expected. Pearson correlation coefficient (two-sided) was used to measure the correlation between mean telomere length and age ($r = -0.252$; $p$-value = 0.023). Source data are provided in Source Data 2. **b** Mean raw telomere length distribution of patient PBL samples shows age-dependent shortening as expected. Pearson correlation coefficient (two-sided) was used to measure the correlation between mean raw telomere length and age ($r = -0.228$; $p$-value = 0.041). Source data are provided in Source Data 2. **c** Bar graph showing the percentage of different G strand end sequences across all 81 patient PBL samples. The most common G strand end observed is 5′-GGTTAG-3′. Source data are provided in Source Data 2.

98% of the ends of 12nt telomeric reads terminate with "GT", which corresponds to the product of RsaI digestion site. Such sequencing reads with only two telomere repeats are very rare (Fig. 2e) and does not significantly affect the overall mean telomere length estimation (Figs. 2d and S4c). Similar results were observed when raw telomere length was used for length estimation and comparison (Fig. S5). In addition, the estimated mean telomere length derived from independent sequencing runs also showed very good concordance (Fig. S6a). The slight variation in mean telomere length observed between independent sequencing runs might be due to differences in loading efficiencies, as a loading bias is known to exist for shorter genomic DNA fragments[47]. To estimate the ideal sequencing depth needed for accurate telomere length measurement, we combined the telomere-containing reads obtained from different biological replicates of HCT116 and 293T cells from independent sequencing runs and performed bootstrap resampling to calculate the standard error (SE) of mean in telomere length measurement. As shown in Fig. S6b and c, our results indicate that a sequencing depth of ~20,000 telomere reads is sufficient to reduce the expected error of mean to less than ±20nt with 95% confidence (1.96*SE). Hence, the sensitivity of this method is sufficient to detect the annual telomere shortening rate of ~43 bp in adult humans[33].

## High throughput telomere length measurement in patient PBLs

To apply PacBio HiFi sequencing for high throughput telomere length measurement, we multiplexed 104 patient PBL samples in two separate runs (Supplementary Data 3). To identify and demultiplex telomere-containing reads for specific patients, we applied our in-house bioinformatics pipeline, Telomap[48]. In general, we obtained at least 2000 telomere-containing HiFi sequencing reads for each patient sample. We calculated the mean telomere length or mean raw telomere length for each patient and examined its correlation with age (Fig. 3a, b). As expected, our results showed a negative correlation between telomere length or raw telomere length with increased age (Pearson's correlation coefficient, SciPy[49] (v1.9.0)). The distributions of telomere lengths and raw telomere lengths in patient PBLs are also shown by violin plot (Fig. S7 and Supplementary Data 4). These results indicate that PacBio HiFi sequencing is a reliable method for assessing telomere lengths at the level of individual telomere with nucleotide resolution.

## G strand processing of human telomere

Our long-read telomere sequencing methodology also allows for accurate profiling of G strands at chromosomal ends. An earlier study carried out in BJ fibroblast cells showed that G strand processing is less precise in mammalian telomeres than in *Tetrahymena* and *Euplotes*[50,51].

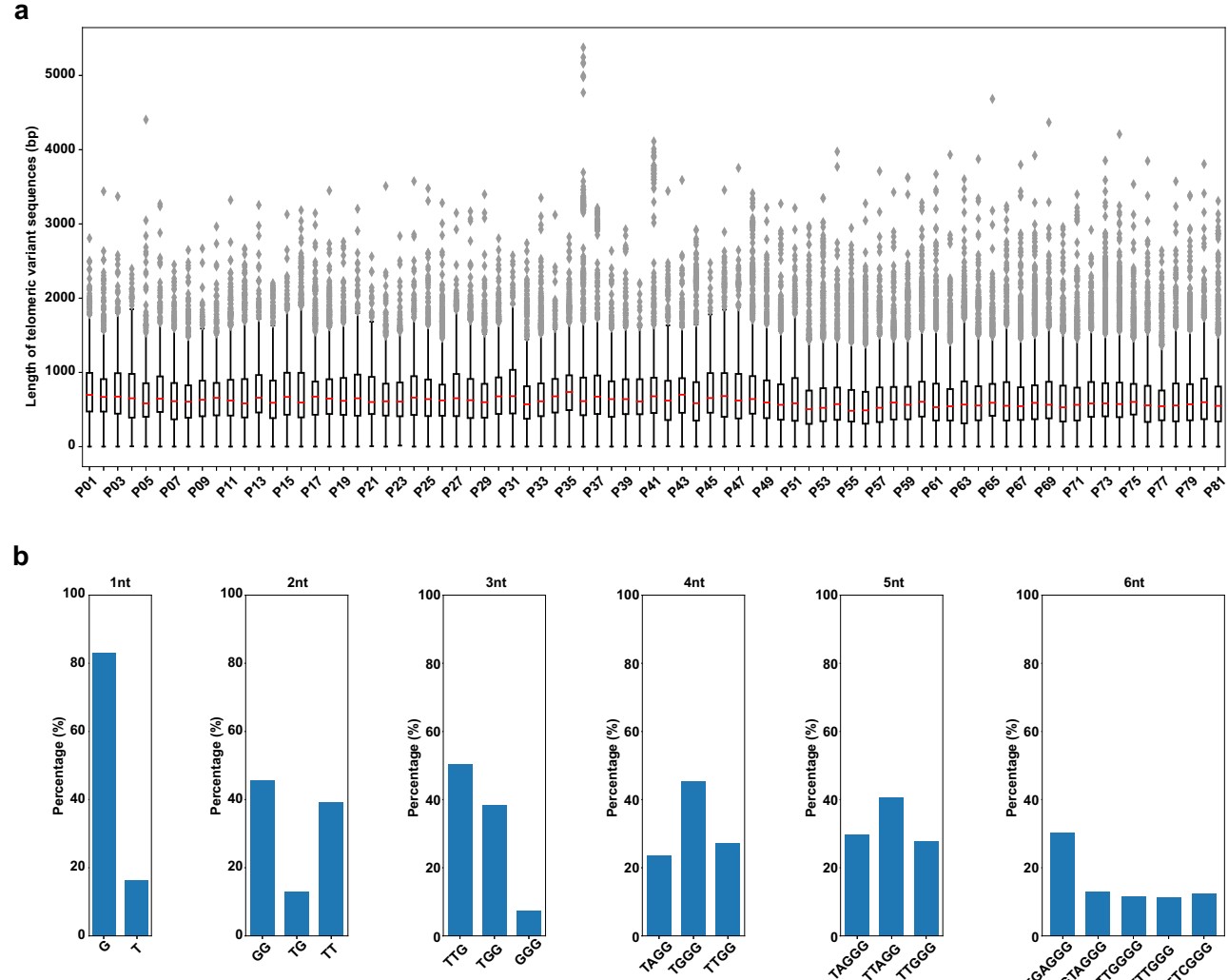

**Fig. 4 | The unique length distribution and sequence of telomeric variant sequences (TVSs) in patient PBL samples. a** Boxplots showing the length distribution of TVSs within the telomere repeat region that is unique to each PBL sample. Please refer to the Source Data 2 file for the *n* value of each sample. Within each box, the red center line denotes the median value, the bounds of box represent the 25th to 75th percentile values, the whiskers represent adjacent values within 1.5 interquartile range, the ends of the whiskers depict the minimum and maximum values within the range, and gray diamonds denote outliers beyond the range. Source data are provided in Source Data 2. **b** The unique sequence preference of TVSs that are 1 to 6 nucleotides in length, compiled from all the telomere repeat-containing HiFi sequencing reads obtained from the 81 patient PBL samples. Only TVSs with proportion >5% are displayed. Source data are provided in Source Data 2.

It was estimated that about 70% of mammalian G strands end in either GGTTAG-3′, GGGTTA-3′, or AGGGTT-3′[51]. Here, our PacBio HiFi sequencing results obtained from patient samples (Fig. 3c) and culture cells (Fig. S8a) indicate that the human telomere G strand processing is very similar to that observed in *Tetrahymena* and *Euplotes*, with more than 60% of the G strands ending in 5′-GGTTAG-3′ (Figs. 3c and S8a). This is to be expected given that the 5′-GGTTAG-3′ sequence is predominantly generated (as specified by the RNA template) during telomerase-dependent telomere elongation.

**Unique distribution and sequence of TVSs in telomeric region**

Consistent with previous publications[42–44], our data revealed the presence of TVSs within the telomere repeat region and they exhibit very high degree of heterogeneity in length and sequence (Fig. 4a). To further investigate their unique features and identify specific sequence preferences, we analyzed TVSs in the patient PBL samples that ranges from one to six nucleotides in length. Our results showed that the sequence distributions of TVSs are not random (Fig. 4b). For example, mono-nucleotide TVSs are dominated by G and T; di-nucleotide TVSs

are dominated by GG, TG, and TT; tri-nucleotide TVSs are dominated by TTG, TGG, and GGG; tetra-nucleotide TVSs are dominated by TAGG, TGGG, and TTGG; penta-nucleotide TVSs are dominated by TAGGG, TTAGG, and TTGGG; and hexa-nucleotide TVSs are dominated by TGAGGG, GTAGGG, TTGGGG, TTTGGG, and TTCGGG. However, the specific implications and functions of these sequence preferences remain to be addressed.

In addition, we found that the abundance and size distribution of TVSs are unique to each culture cell line and patient sample (Figs. 5a, S8b, and S9). Furthermore, the unique features of TVSs are not limited to the Singapore cohort (81 samples, mainly Chinese), but are also present in African American (4), Hispanic (6) and Caucasian (12) PBMC samples. To probe the distribution of TVSs in the telomeric region, we constructed the TVS signature plot for specific telomere ends of each patient sample. The TVS signature presents the distribution of TVS by showing the percentage of TVS per nucleotide across all telomeric reads of a given end. It begins from the 5′ start of the telomere and spans 3 kb in length toward the 3′ end. The TVS percentage indicates the proportion of telomeric reads that contains

TVSs at a specific nucleotide position. A TVS percentage of 100% indicates that TVSs are present in all telomeric reads, whereas 0% indicates that canonical telomeric repeats are present in all telomeric reads at a given position. K-means[52] clustering was also performed to classify each telomeric read into potential haplotypes, which were then balanced in numbers. Using Chr. 7q as an example, the TVS signature of haplotype-balanced telomeric reads show the predominant presentation of larger and more abundant TVSs near the sub-telomeric end of the telomere repeat region (Figs. 5b and S10). In addition, we demonstrated that the signature of TVS distribution in individual mappable chromosome ends (such as Chr. 7q, 2p, 3p, 5p, 6q, 11q, 12q, 14q, and 17p) is unique for each patient sample that was compared by calculating their Euclidean distance (Figs. 5c, S11, and Supplementary Data 5). Moreover, high concordance in TVS signature correlation is observed in biological replicates of cultured cell lines as well as patient samples re-sequenced in independent sequencing runs (Figs. S12, S13, and Supplementary Data 1).

### TVSs disrupt the binding of shelterin complex at telomere

Previous studies[53,54] and our results (Fig. S14) show that mutations in the canonical telomere sequence dramatically reduced the binding affinity of TRF1 and TRF2 and hence impaired chromosomal end protection. Therefore, the presence of larger and more abundant TVSs near the sub-telomeric region predicts a lower shelterin binding density when the telomeres become very short, which may lead to precipitative deterioration of telomere protection later in life. To address such a possibility, we calculated the maximum number of TRF1/TRF2 binding motifs within the first 1 kb of canonical telomere repeats next to the sub-telomeric region for all chromosomal ends in each patient sample. Since the length of the shortest telomeres, rather than the average telomere length, is known to better predict the viability and long-term proliferative capacity of tissue stem cells[26], we focused on the bottom 5% of telomeres that have the least number of TRF1/TRF2 binding sites within this 1 kb region for each patient sample. As shown in Fig. 5d, we observed dramatic differences in TRF1/TRF2 binding densities in this region in our patient samples. To further illustrate our observation, we focused on the ends of individual chromosome, such as Chr. 7q from patient samples P14 and P15. We noted that both chromosomal ends have different raw telomere lengths (1126 bp vs 1627 bp) and number of TRF1/TRF2 bindings sites (81 vs 62) due to the presence of TVSs, despite having an identical length of canonical telomere repeats of 1 kb (Figs. 5e, S15 and S16). The reduced TRF1/TRF2 binding density, even at only one chromosome end, predicts a lower telomere protection by the shelterin complexes, thereby increasing the probability of telomere uncapping and the induction of DNA damage responses in the event that the telomeres become very short. This increased risk of telomere uncapping due to the presence of TVSs may have profound negative implications in human aging and diseases, especially later in life[9–12].

## Discussion

In this study, we have presented a method for the enrichment of telomere-containing genomic DNA fragments for single-molecule real-time sequencing using the PacBio HiFi sequencing platform. We have validated the possibility of using this method for measuring telomere lengths at nucleotide resolution, with accuracy and consistency that is pertinent to epidemiological studies as well as the longitudinal assessment of telomere attrition in individuals. Furthermore, our method enables the unbiased mapping of extremely short telomeres at the chromosomal ends with better sensitivity than STELA and its variants. Consistent with previous observations[42], our method has identified the over-representation of extremely short telomeres in T24 cells.

The trimming of telomere-containing genomic DNA is necessary for telomere length measurement using PacBio HiFi sequencing as the maximum read length is limited by the processivity of the DNA polymerase. We have validated the use of HinfI and RsaI for trimming of genomic DNA prior to the enrichment of telomere-containing DNA fragments using telobaits. However, one potential drawback of trimming using RsaI and HinfI is that the resulting subtelomeric region from the HiFi reads could be too short for unambiguous identification of specific chromosome ends with reference to T2T-CHM13 human genome assembly[55]. A careful selection of restriction enzymes for the purpose of genomic DNA trimming is necessary. For example, we found that HphI and MnlI restriction endonucleases, which are known to cut closer to the telomere-containing region, frequently cut in the TVSs within the telomere repeat regions (Fig. S3). This will result in the loss of telomere repeat regions closer to the sub-telomeric end during the DNA fragment enrichment process, leading to the underestimation of telomere length. Another caveat of this method is the incompatibility of using PacBio HiFi sequencing reads in measuring telomere length of ALT cells. ALT cells have highly heterogenous telomere length distribution with very long telomeres (Fig. 1b) that are beyond the optimal processivity of PacBio HiFi DNA polymerase.

We have also shown that the telomere repeat region is not homogenous but is interspersed with TVSs of unique sizes and sequences. These unique features were not bounded by ethnicity but rather specific to individuals. With more validations and fine-tuning, this unique feature might potentially be utilized as a fingerprint to identify cell lines or human samples in the future. The presence and distribution of TVSs within the telomere repeat regions are likely due to errors during DNA replication, which are inherited in the ensuing cell progeny. They may be used to trace the replication history of a cell as new mutations are likely to be incorporated over time within the individual's lifespan. We observed the presence of larger and more abundant TVSs within the telomere repeat regions near the sub-telomeric end. This feature is not specific to telomeres at any individual chromosomal ends but is detectable at most chromosomal ends (Figs. 5b and S10). Telomere elongation by the error-prone hTert enzyme may also contribute to the generation of these TVSs[56], which may explain the predominance of small TVSs near chromosome ends.

As the shelterin complex (specifically the TRF1 and TRF2 subunits) binds specifically to the canonical telomere sequence motif in the double-stranded DNA, the presence of larger and more abundant TVSs near the sub-telomeric region predicts a poorer telomere protection later in life, when telomeres become very short (Fig. 5d, e). This implies that not only the absolute telomere length, but also the continuity of canonical telomere repeats are key parameters in predicting lifespan and aging-related diseases in the general human population[22]. In the future, the ability to map each telomere-containing reads to a specific chromosomal end could help us to study the haplotype diversity of telomere distribution in the general population as well as in individuals with telomere diseases who inherited allele-specific telomeres from their parents[57].

Our established method for telomere length measurement provides a high-throughput (up to 72 patient samples in a single run so far) and accurate estimation of the absolute telomere length at individual chromosomal end. The nucleotide-level resolution achieved by this method will be essential for longitudinal assessment of telomere attrition[33] in individuals within a short period of time. While previous epidemiological studies using qPCR-based telomere length measurement relied on a large population base for statistical compensation[25], our established method will allow for the interpretation of phenotypes in a small and controlled study settings (such as clinical trials) to evaluate the effects of potential interventions on the steady-state telomere maintenance in patients. The sensitivity of this method to detect small changes in telomere length could make it an essential tool not only for estimating the trajectory of changes in telomere length that may function as an important biomarker, but also for developing potential interventions that can slow down population aging in the future.

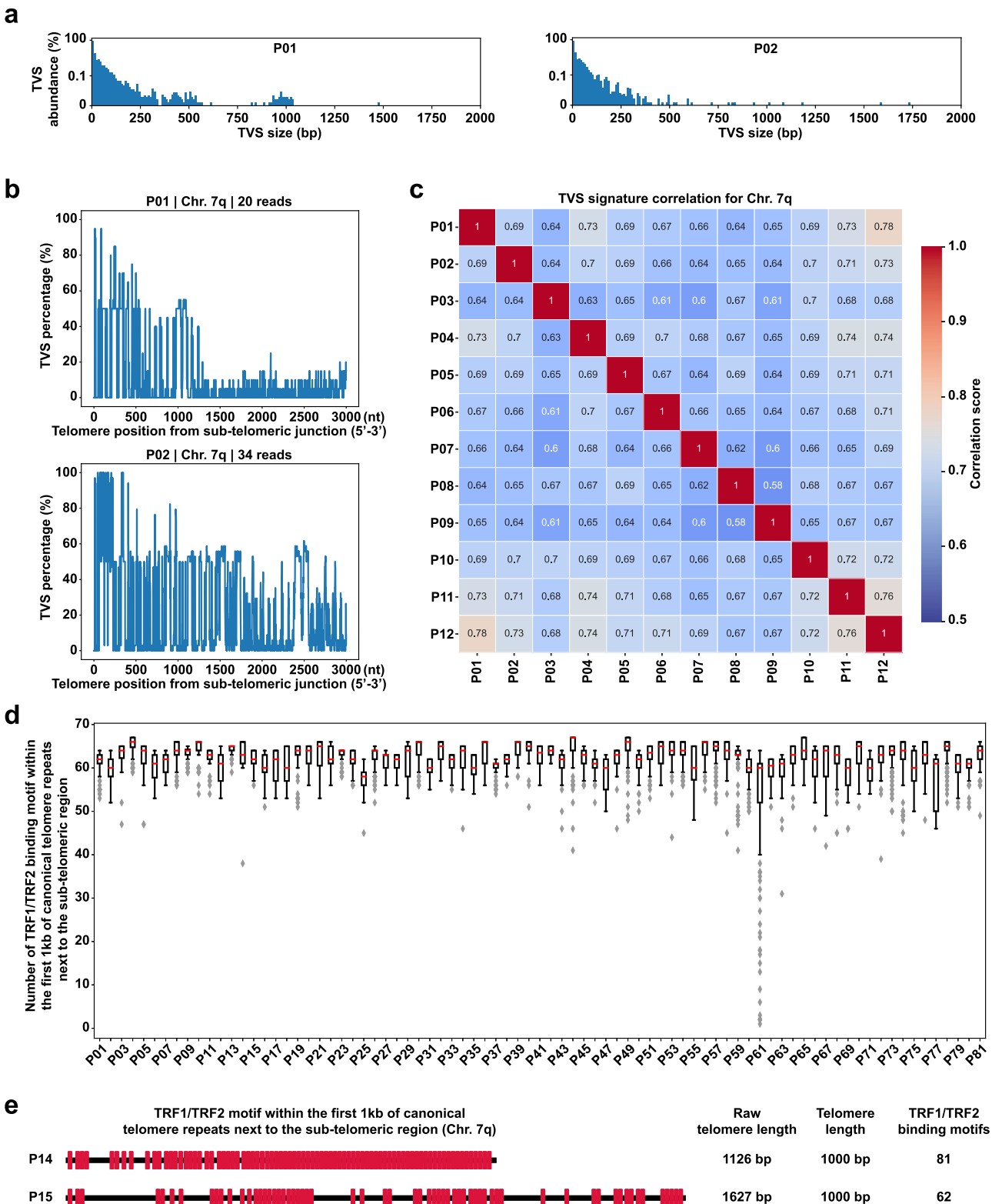

## Methods

### Ethical statement

All experiments comply with relevant ethical regulations approved by the Ethics Committee of the National University of Singapore. The use of human biological materials for research was approved by National University of Singapore Institution Review Board and SingHealth Centralised Institution Review Board.

### Cell lines and plasmid

Commercial cell lines–HCT116, 293T, HeLa, T24, VA13, U2OS, IMR90 and WI38 (purchased from ATCC) were cultured in DMEM with 10% FBS, and 1% Pen-Strep. Cells were sub-cultured at a ratio of 1:8 and seeded at a density of $5 \times 10^5$ in a 10 cm dish for the experiments. WA18 ES cells (purchased from WiCell) were cultured in mTeSR1. Cell authentication was performed by short tandem repeat (STR) profiling. pWY82 vector was purchased from Addgene (#65721).

**Fig. 5 | Presence of TVSs may uniquely affect chromosome end protection for each individual. a** Histograms showing the unique TVS size abundance profile of P01 and P02. The Y-axes are presented in logarithmic scale. Only TVSs with sizes ≤2 kb are shown. **b** TVS signature plots of samples P01 and P02 for telomeres at Chr. 7q. Longer and more abundant TVSs are present near the sub-telomeric ends of telomere repeat regions in the patient PBL samples as shown by an increase in TVS percentage. **c** The TVS signature correlation of Chr. 7q for P01 to P12. TVS signatures were pairwise compared by Euclidean distance. **d** Boxplots showing the density distribution of the TRF1/TRF2 binding motif on the bottom 5% of telomeres with the fewest binding sites in each sample (P01-P81). Only binding motifs occurring within the first 1 kb of canonical telomere repeats next to the sub- telomeric region were counted. Source data are provided in Source Data 2. Please refer to the Source Data file for the n value of each sample. Within each box, the red center line denotes the median value, the bounds of box represent the 25th to 75th percentile values, the whiskers represent adjacent values within 1.5 interquartile range, the ends of the whiskers depict the minimum and maximum values within the range, and gray diamonds denote outliers beyond the range. **e** An example of differential TRF1/TRF2 binding densities on telomeric reads at Chr. 7q from P14 and P15 PBL samples. The black lines represent telomeres starting from the 5′ end (next to the sub-telomeric region) until the end of the first 1 kb of canonical telomere repeats. The red markers represent the TRF1/TRF2 binding motifs. Source Data 2.

## Human samples

The subjects were recruited from the Cardiac Ageing Study (CAS)[58], a prospective study that examines characteristics and determinants of cardiovascular function in older adults. Written informed consent was obtained from participants upon enrollment. The SingHealth Centralised Institutional Review Board had approved the study protocol (CIRC/2014/628/C).

Antecubital venous blood samples were taken from consenting participants and stored in PAXgene blood DNA tube (BD Biosciences). We used fresh or archived samples in the current study. For fresh samples, after collection, the blood samples were immediately processed. For archived samples, after collection, the blood samples were immediately placed on ice for transportation and were processed within 6 h to obtain buffy coat samples, which were subsequently stored at −80 °C. PBL genomic DNA was purified using PAXgene blood DNA kit (Qiagen).

## Genomic DNA extraction and Southern blotting (Teloblot)

Genomic DNA from cultured cells was extracted using Gentra Puregene genomic DNA purification kit (Qiagen). Telomere Southern hybridization was done as previously described[59]. In brief, 0.5 µg of genomic DNA was digested with RsaI/HinfI or EcoRI as indicated at 37 °C for 16 h. The digested DNA was normalized and run on 0.6% agarose gel (Seakem ME agarose, Lonza) in 1× TBE. After the run, the agarose gel was first depurinated in 0.25 M HCl for 30 min with gentle shaking at room temperature, followed by denaturation in 0.5 M NaOH and 1.5 M NaCl for 30 min with gentle shaking at room temperature, and finally incubated in neutralization buffer (1 M Tris pH 7.4, 1.5 M NaCl) for 30 min with gentle shaking at room temperature. The gel was then set up for capillary transfer using Whatman gel blot paper (GB003, GE Healthcare) to transfer the digested genomic DNA onto Hybond-XL membrane (GE Healthcare) in 10x SSC buffer. After overnight transfer, DNA blot was first crosslinked using XL-1500 UV crosslinker (Krackeler Scientific Inc.), then prehybridized in phosphate buffer (0.5 M NaPO4, pH 7.2, 7% SDS, 1 mM EDTA, pH8.0) for 1 h with rotation in hybridization chamber at 37 °C. The $^{32}$P-labeled (TTAGGG)$_6$ oligonucleotides were then added and incubated with the DNA blot overnight with rotation in hybridization chamber at 37 °C. The DNA blot was then washed in phosphate wash buffer (0.2 M NaPO4 pH7.2, 1%SDS,1 mM EDTA) for 15 min with rotation in hybridization chamber at 25 °C for a total of three times before exposing on a phosphorimager screen.

## Synthesis of barcoded Telobaits

The oligos (Supplementary Data 2) for telobaits were synthesized by Integrated DNA Technologies. The forward primers with 3′ biotin label were synthesized at 100 nM scale and further purified by HPLC. The six reverse primers, each containing a single telomere repeat with all six possible ends that can anneal to the telomeric ends, were synthesized at 25 nM scale and purified by desalting. The oligo mix was heated at 95 °C for 10 min and then slowly cooled down to room temperature to facilitate the annealing of forward and reverse primers (at a ratio of 1:6). The

concentration of the telobaits was further adjusted before telomere enrichment.

## Telomere enrichment using telobaits

For telomere enrichment in genomic DNA extracted from culture cell lines or patient samples, the genomic DNA was first quantified using Qubit dsDNA BR assay kit (Thermo Fisher Scientific). Following this, about 20 µg of genomic DNA was incubated with 5 µl of 10 µM of telobaits (molar ratio of 10,000:1), 5 µl of 10 mM ATP, 2 µl of T4 DNA ligase (30 U/µl), 2 µl each of FastDigest RsaI and HinfI in 1x FastDigest buffer (Thermo Fisher Scientific) at a total volume of 100 µl in 1.5-ml Eppendorf DNA LoBind tubes. The tubes were incubated overnight at 37 °C with shaking at 1000 RPM on ThermoMixer C with attached ThermoTop. The next day, 150 µl of Dynabeads MyOne Streptavidin T1 beads were first washed with 2× binding and washing solution (10 mM Tris, pH7.5; 1 mM EDTA and 2 M NaCl) and then added to the ligation mix. The beads and the ligation products were incubated at room temperature for 2 h with shaking at 1000 RPM on ThermoMixer C. Next, the beads were washed with 1× binding and washing solution (5 mM Tris, pH7.5; 0.5 mM EDTA and 1 M NaCl) three times with rotation at room temperature for 5 min. The beads were then equilibrated with 1× FastDigest buffer at room temperature for 5 min before filling the gaps using T4 DNA polymerase: 1 µl of T4 DNA polymerase (5U/µl), 1 mM dNTP, in 1× FastDigest buffer at a total volume of 50 µl in 1.5-ml Eppendorf DNA LoBind tubes. The gap filling using T4 DNA polymerase was done at 37 °C for 30 min with shaking at 1000 RPM on ThermoMixer C with attached ThermoTop. The beads were then washed with 1× binding and washing solution (5 mM Tris, pH7.5; 0.5 mM EDTA and 1 M NaCl) three times with rotation at room temperature for 5 min. The beads were then equilibrated with 1× FastDigest buffer at room temperature for 5 min and subsequently digested with EcoRI enzyme to release the telomere genomic DNA fragment from Dynabeads MyOne Streptavidin T1 beads. The genomic DNA fragment was then purified using 0.45× (volume/volume) AMPure PB beads (PacBio) according to the manufacturer's protocol. Finally, the library was then engineered using SMRTbell Express Template Prep Kit 2.0 (PacBio) as per manufacturer's recommendation.

## Gel mobility assay and quantification of telomere dysfunction-induced DNA damage foci (TIFs)

Gel mobility assay for TRF1 and TRF2 was done following previous published protocol[42]. For TIFs analysis, WA18 hTR$^{(−/−)}$ cells were infected with lentivirus co-expressing the telomerase template mutants and a puromycin-resistance gene. Two days post-infection, all cells (except for TGAGGG) were treated with 1 µg/ml puromycin for 48 h to eliminate uninfected cells. Immunostaining was carried out 5 days post-infection. For TGAGGG, we infected the cells and carried out immunostaining 3 days post-infection (skipping the puromycin selection) as this template mutant was particularly detrimental to cell proliferation and all infected cells died 5 days post-infection. Immunofluorescence staining and subsequent image acquisition/analysis were carried out as previously described[60]. Briefly, cells grown on coverslips were fixed with 2% paraformaldehyde and permeabilized with 0.5% NP-40.

Immunofluorescence staining was carried out by incubating with a rabbit anti-53BP1 antibody (Novus Biologicals, NB100-304, 1:500) and a mouse anti-TRF1 antibody (Novus Biologicals, H00007013-B01, 1:500), followed by secondary antibodies Donkey anti-Mouse IgG (H + L) Highly Cross-Adsorbed Secondary Antibody, Alexa Fluor™ 488 (ThermoFisher Scientific, A-21202, 1:500) and Donkey anti-Rabbit IgG (H + L) Highly Cross-Adsorbed Secondary Antibody, Alexa Fluor™ 568 (ThermoFisher Scientific, A10042, 1:500) respectively. Cell images were acquired using a Nikon Ti-U microscope with a 100x objective and collected as a stack of 0.2 μm increments in the z-axis. After deconvolution using the AutoQuant X3 software (Media Cybernetics), images were viewed with the maximal projection option on the z-axis. All image files were randomly assigned coded names to allow blinded scoring for spots colocalization. Two independent experiments were performed.

## Analysis of Telobait-enriched PacBio HiFi sequencing data

The bioinformatics pipeline described here is compiled into Telomap[48] [https://github.com/cytham/telomap], a command-line program written in Python. Long reads generated from PacBio HiFi sequencing were first parsed by Pysam (v0.19.1), a wrapper around HTSlib[61] and SAMtools[62], and then queried for the capture oligo sequence of Telobaits using the "Bio.Align" package from Biopython[63]. The capture oligo sequence search was limited to the last 150 bases at the 3′ end of either read strand and required a perfect alignment score. Next, the reads were de-multiplexed by querying for sample barcode sequences, which also needed a perfect alignment score. Sample de-multiplexing information can be found in Supplementary Data 1. To remove read artifacts generated during library preparation, reads containing more than one capture oligo sequence or sample barcode sequences were omitted from the analysis. The remaining valid reads were briefly queried for the presence of at least two canonical telomeric 5′-TTAGGG-3′ repeats within 50 bp adjacent to the capture oligo site using the "re.finditer" function from the Python standard package library[64]. Subsequently, reads containing the repeats were extensively queried for all 5′-TTAGGG-3′ repeats throughout the entire read. The telomeric region of each read was then defined to begin at the 5′ end with the first occurrence of a tandem telomeric repeat (5′-TTAGGGTTAGGG-3′) and conclude at the 3′ end before the start of the capture oligo. Non-canonical telomeric sequences found within the telomeric region were defined as telomeric variant sequences (TVSs), and their sizes, sequences, and abundances were evaluated. The other non-telomeric sequence region of each read, which starts from the 5′ end to the onset of the telomeric region, was presumed as the sub-telomeric region. The sub-telomeric region of all reads were aligned to one another and clustered into groups as candidate chromosomal ends. The clustering was performed using the "cluster.py" module of Telomap with a minimum sequence identity of 95% and a minimum length of 100 bp. Within each candidate chromosomal end group, a 100 bp consensus sequence was generated from the most common bases across reads. These consensus sequences were mapped to the sub-telomeric regions (1 kb) of the T2T-CHM13 genome[55] to identify the chromosomal ends of each group. All analyzed data were organized in a Pandas[65] (v1.4.3) dataframe and plots were generated using Seaborn[66] (v0.11.2) and Matplotlib[67] (v3.5.3) packages. Information on all four PacBio sequencing runs such as ZMW metrics, CCS coverage, and CCS read length can be found in Supplementary Data 1 and Fig. S17. Biological replicates of culture cell lines and patient samples were assayed to ensure the consistency of the analysis.

## Analysis of linearized DNA vector control

The repetitive 5′-TTTAGGG-3′ (1121 bp) and non-repetitive (8549 bp) sections of the linearized DNA vector were analyzed separately. PacBio HiFi reads were first identified as the DNA vector by aligning to the repetitive or non-repetitive reference sequences using HS-BLASTN v2.0.0[68] with the "-outfmt 6 -max_target_seqs 10" parameters, where a minimum span of 95% in length across the reference is required. The HS-BLASTN output also reported the alignment accuracy of each read for the respective sections. The sequencing quality of the reads was obtained directly from the PacBio HiFi output BAM file and converted into Phred quality scores. Reads with at least Q20 were used in the analysis of alignment accuracy and sequencing error. The sequencing error of each read was deciphered from the CIGAR string in the BAM file after read alignment using Pbmm2 v1.4.0 (https://github.com/PacificBiosciences/pbmm2) with the "CCS" preset parameter. The errors consisted of insertions, deletions, and mismatches in relation to the reference sequence. Only errors with a relative frequency ≥0.01 amongst all vector reads were reported.

## Bootstrap resampling

The bootstrap method was used as the resampling technique to estimate the mean telomere length in cells, as well as the standard error (SE) of the mean across different sample sizes or telomere sequencing depths. Resampling was performed on the telomere reads from the HCT116 (25,320 reads) and 293T (24,785 reads) cell lines with replacement. The sample sizes defined were 1000, 2000, 5000, 10,000, 15,000, and 20,000 reads, and random sampling was repeated for 10,000 iterations for each sample size. The SE of mean ($\theta$) for each sample size ($B$) was estimated by

$$SE(\theta) = \sqrt{\frac{1}{B}\sum_{b=1}^{B}\left(X^b - \frac{1}{B}\sum_{i=1}^{B}X^i\right)^2} \quad (1)$$

which is the standard deviation of the bootstrap mean ($X$). The 95% confidence interval was calculated using the critical value of 1.96, where the intervals correspond to $\theta \pm 1.96 \cdot SE$. Bootstrapping was performed in Python, using the "random" package for random sampling with replacement and the "statistics" package for calculation of mean and standard deviation[64]

## TVS signature analysis

TVS signature plots visualizes the proportion and location of TVSs on telomeres at specific chromosomal ends. TVSs are defined to be any sequences apart from the canonical telomere repeat 5′-TTAGGG-3′ within the telomeric region of a sequencing read. The Y-axis of a TVS signature plot denotes the TVS percentage, which reflects the proportion of TVS at each nucleotide position. The X-axis denotes the 5′ to 3′ telomeric nucleotide position which begins from the 5′ start of the telomere and spans a length of 3 kb toward the 3′ end. For a given telomere, which corresponds to the telomeric region of a telomeric read, we assign a value of 0 to all bases belonging to the canonical telomeric repeat, and a value of 1 to the remaining bases, which are TVSs. A 2D matrix can be produced using NumPy v1.20.3[69] by aggregating all telomeres at a specific chromosomal end, with the rows representing each telomere and the columns representing each base position in the 5′−3′ direction. To fill any empty data points caused by telomere length discrepancies, a placeholder value of 0 was assigned. To establish a balanced representation of two haplotypes, K-means clustering (Scikit-learn[52] (v1.1.2)) was conducted on the 2D matrix to attempt classifying the telomeres into two groups ($k = 2$). Next, the 2D matrix is balanced to contain the same number of telomeres from each cluster by down-sampling the larger cluster to the size of the smaller cluster. Clusters are required to be at least 25% in size relative to the total number of telomeres, otherwise, the haplotype balancing step will be omitted. Finally, the TVS percentage for each base position can be calculated by the summation of data points column-wise, then dividing by the number of telomeres and

converting to a fraction of 100, as denoted by

$$Y_i = 100 \cdot \frac{1}{N} \sum_{t=1}^{N} R_t^i \qquad (2)$$

where $Y_i$ is the TVS percentage at position $i$, $N$ is the total number of telomeres at a specific chromosomal end, and $R_t^i$ is the assigned value (0 or 1) at position $i$ for telomere $t$. A region with 0% TVS means that canonical telomeric repeats are consistently present at that region across all reads of a specific chromosomal end, whereas a region with 100% TVS means that TVS are consistently present at that region across all reads of a specific chromosomal end.

## TVS correlation calculation

The comparison of two TVS signatures is performed by measuring their Euclidean distance[49], $d$, which is defined as

$$d(x,y) = \sqrt{\sum_{i=1}^{n}(x_i - y_i)^2} \qquad (3)$$

where $x$, $y$ are two TVS 1D arrays, $x_i$, $y_i$ are respective TVS percentages at position $i$, and $n$ is the length of the array (3000 bp). To put the Euclidean distance into perspective, we divide it by the maximum Euclidean distance attainable for $n = 3000$ with data values of 0 and 100. The final pairwise TVS correlation score between two TVS signatures is calculated as

$$s(x,y) = 1 - \frac{\sqrt{\sum_{i=1}^{n}(x_i - y_i)^2}}{\max(d_n)} \qquad (4)$$

where $s$ is the TVS correlation score that ranges from [0, 1], and max $(d_n)$ is the maximum Euclidean distance for $n$. A high TVS correlation score implies high TVS signature similarity. In our study, the TVS correlation score between biological replicates is >0.90 (Figs. S12 and S13b).

## TRF1/TRF2 binding motif analysis

The positions of TRF1/TRF2 binding motif 5′-TTAGGGTTA-3′ on each telomeric read was identified using the "re.finditer" function from the Python standard package library[64]. To control for discrepancies in length between telomeres, the number of TRF1/TRF2 binding motifs was normalized by telomere length defined by canonical telomeric repeats. Only motifs found within the first 1 kb of canonical telomeric repeats were considered.

## Statistics and reproducibility

Telomere length analyses was performed in five culture cell-lines and 104 patient samples across five different ethnicities. Four biological replicates of HCT116 and 293T, three replicates of T24, and two replicates of IMR90 and WI38 cell-lines were used to assess the accuracy of telomere length measurement using PacBio HiFi sequencing. The replicability of TVS signature was evaluated using biological replicates from independent sequencing runs of 12 patient samples. Statistical analyses were performed using the NumPy (v1.20.3)[69] and SciPy[49] (v1.9.0) packages in Python 3 (https://www.python.org/). No statistical method was used to predetermine sample size and no data were excluded from the analyses. For the study of telomere dysfunction-induced DNA damage foci (TIF) by TVSs, about 16 images (8–15 cells per image) from two independent experiments were taken randomly, and each image file was assigned coded names to allow blinded scoring of co-localization spots. In sum, more than 120 cells per line were analyzed for TIFs.

## Data availability

The PacBio HiFi sequencing data of patient samples generated in this study have been deposited in the European Genome-phenome Archive (EGA), which is hosted by the EBI and the CRG, under accession codes EGAS00001006103 and EGAS00001006595. The patient sequencing data are available under restricted access for privacy reasons. Access can be granted for institutional researchers for general research use and scientific publication by contacting Shang Li (shang.li@duke-nus.edu.sg) or the Data Access Committee (DAC) [https://ega-archive.org/dacs/EGAC00001002582]. The timeframe of responding to a data access request is two weeks. The PacBio HiFi sequencing data of cultured cell lines are available in the European Nucleotide Archive (ENA) under accession number PRJEB49668. The data used to generate the figures in this study are provided in the Source Data files. Source data are provided with this paper.

## Code availability

The bioinformatics pipeline described here is compiled into Telomap[48] (v0.0.4) [https://github.com/cytham/telomap], a command-line program written in Python.

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

## Acknowledgements

This work was supported by grants from the National Medical Research Council Singapore (NMRC/CIRG/1481/2017) and the Ministry of Education (MOE2017-T2-1-105) to S.L.; a grant from National Medical Research Council Singapore (NMRC/OFIRG18may-0013) to H.W.S.; and a grant from National Medical Research Council Singapore (MOH-000153) to G.S.L. and A.S.K. We thank Hui Qi and Jae Peck Yean Tan from Research Instruments Pte Ltd for their assistance in PacBio HiFi Sequencing. We thank Lei Luo and John Zhang from NovogeneAIT for their assistance in PacBio HiFi Sequencing and data analysis. We thank Dr. Ravinuthula Sruthi Jagannathan for reading the manuscript.

## Author contributions

C.Y.T. performed long read sequencing data analysis. L.P., T.D.Y., J.Y.P.K., and M.K.R. performed genomic DNA purification and telomere capture for PacBio HiFi sequencing. V.S.I.T. and Y.C. performed data analysis for telomere length measurement in cancer cell lines. S.Z. and L.X. performed immunostaining presented in this study and analyzed the data. Z.H. and H.W.S. purified protein used in this study. G.S.L. and A.S.K. collected the patient samples used in this study. J.L. B.T.T., and P.T. interpreted the results and provided comments to the manuscript. M.O. and S.L. designed the study. All authors contributed to the preparation of the paper.

## Competing interests

The authors declare no competing interests.

## Additional information

[1]Cancer Science Institute of Singapore, National University of Singapore, 14 Medical Drive, Singapore 117599, Singapore. [2]Cancer and Stem Cell Biology Program, Duke-NUS Medical School, 8 College Road, Singapore 169857, Singapore. [3]School of Life Sciences, Shanghai University, 99 Shangda Road, Shanghai 200444, China. [4]Department of Microbiology and Molecular Genetics, University of California, Davis, CA 95616, USA. [5]Key Laboratory of Molecular Target & Clinical Pharmacology, School of Pharmaceutical Science, Guangzhou Medical University, Guangzhou 511436, China. [6]Institute of Molecular and Cell Biology, Agency for Science, Technology and Research, (A*STAR), 61 Biopolis Drive, Singapore 138673, Singapore. [7]National Heart Centre Singapore, Duke-NUS Medical School, 5 Hospital Drive, Singapore 169609, Singapore. [8]Centre for Quantitative Medicine, Duke-NUS Medical School, 8 College Road, Singapore 169857, Singapore. [9]School of Data Science, The Chinese University of Hong Kong-Shenzhen, 2001 Longxiang RoadLonggang District Shenzhen 518172, China. [10]Department of Haematology, Singapore General Hospital, 1 Hospital Drive, Singapore 169608, Singapore. [11]Hematopoietic Stem Cell and Cellular Therapy Laboratory, Division of Medical Sciences, National Cancer Centre Singapore, 11 Hospital Drive, Singapore 169610, Singapore. [12]Laboratory of Cancer Epigenome, Division of Medical Science, National Cancer Centre, 11 Hospital Drive, Singapore 169610, Singapore. [13]SingHealth/Duke-NUS Institute of Precision Medicine, National Heart Centre Singapore, Singapore 168752, Singapore. [14]Epigenetic and Epitranscriptomic Regulation Domain, Genome Institute of Singapore, Agency for Science, Technology and Research, (A*STAR), 60 Biopolis Drive, Singapore 138672, Singapore. [15]International Research Center for Medical Sciences, Kumamoto University, 2-2-1 Honjo, Chuo-ku, Kumamoto 860-0811, Japan. [16]Department of Physiology, Yong Loo Lin School of Medicine, National University of Singapore, 2 Medical Drive, Singapore 117597, Singapore. [17]These authors contributed equally: Cheng-Yong Tham, LaiFong Poon. ✉e-mail: motomi.osato@gmail.com; shang.li@duke-nus.edu.sg

