## [Transparent Peer Review File · Nature Communications]

High-throughput telomere length measurement at nucleotide resolution using the PacBio high fidelity sequencing platformREVIEWER COMMENTS

Reviewer #1 (Remarks to the Author: Overall significance):

The study by Tham et al., presents a high-throughput method for obtaining accurate telomere length measurements at single-nucleotide resolution. By leveraging highly accurate single-molecule PCR-free sequencing, this method overcomes many of the limitations associated with prior methods for determining telomere lengths, and the enrichment strategy appears well suited for multiplexing multiple samples in the same run. Overall, the method and analyses regarding telomeric variant sequences (TVSs) significantly advance the field and will likely be of general interest. However, the current manuscript contains several conclusions that are not well supported by their data, and additional discussion is needed regarding the limitations of this approach in terms of recovering accurate telomere lengths and telomere lengths specific to a given genomic location.

Major comments:

The abstract claims that “the unique distribution pattern and sequence of TVSs in individuals may be used as a fingerprint for sample identification.” This is a bold claim with wide ranging implications across telomere biology, genetics, and forensics, and has the potential to be picked up in the popular press. However, no data is presented in the manuscript that substantiates this claim. Specifically, although the authors present data that TVSs overall appear to be different across the 48 patient PBMC samples, they do not actually show that if one were to repeat this method with a biological replicate of one of these samples, that it would uniquely match to the same sample. Similarly, as this study is limited to only 48 individuals, it is unclear if the authors are merely identifying various telomere haplotypes that are specific to certain populations, or genetic features that are truly unique to each individual. Please adjust this language accordingly, or substantiate this claim by providing reproducible population-level data across biological replicates from diverse cohorts that accounts for haplotype-specific features.

The abstract claims that “the presence of TVSs disrupts the continuity of the canonical (5'-TTAGGG-3')_n telomere repeats, which affects the binding of shelterin complexes at the chromosomal ends and telomere protection.” However, the authors do not actually demonstrate that TVSs disrupt the binding of shelterin complexes at the chromosomal ends or disrupt telomere protection. Rather, the authors show that TVSs disrupt canonical shelterin binding elements. However, the impact of this on shelterin occupancy and telomere protection is not evaluated in this manuscript.

The calculation of “telomere length” presented in figure 2A is a little awkward and not entirely consistent with standard definitions of telomere length. Specifically, telomere length is often thought of as the continuous linear length of DNA sequence from the start of the TTAGGG repetitive element to the terminal end of a telomere. It is unclear why the authors have chosen to use a non-continuous length measurement. It would be useful for the authors to discuss how their results could differ if they used a continuous linear length definition of telomere length (i.e., “raw telomere length”).

Although this method provides “telomere length measurement at nucleotide resolution”, the authors have not demonstrated that this method can substantially determine which specific telomere a given read arises from. Specifically, the authors only report genomic-location specific results for 15q and 19q (figure 5B), with all other analyses being agnostic of specific genomic location. I suspect that this is at least partially due to the use of hg38 as the reference genome, as telomeric regions within hg38 are quite problematic. However, the lack of genomic position-precise maps beyond 15q and 19q limits the ability to determine the true extent of TVSs within an individual, as it is unclear based on the findings whether these TVSs are fixed within an individual on a given telomere end or are variable across different cells from the same individual. This latter point would have significant implications for using TVSs as a “fingerprinting method”.

For the reads with only a 12nt telomere length, can the authors address whether the ends of these reads happen to correspond to digestion sites for RsaI, HinfI or EcoRI within TVSs, as this would indicate a potential artifactual mechanism for these reads? Similarly, do these reads from the same individual have similar sub-telomeric sequences, which would suggest an off target restriction digestion issue as opposed to a shearing issue as is proposed by the authors.

Minor comments:

The analysis of G-strand ends does not account for the contribution of potential differences in the ligation efficiencies of the 6 telobaits.

If available, the ethnicity of the 48 individuals should be reported to aid in future studies integrating this data.

Please add additional sequencing details such as:

- (1) Number of ZMWs with sequence information from the 48-sample multiplexed run (based on figure 3A, it appears that only ~250,000 reads were recovered, was the SMRT cell underloaded, or were there a lot of reads that failed to get CCS coverage?);
- (2) Per-read CCS coverage versus CCS read length plot from the aforementioned run;
- (3) Demultiplexing efficiency from the aforementioned run (i.e., how many reads were demultiplexed versus those that were unable to be demultiplexed).

Figure S1C, what is the etiology of those three spikes in the histogram on the right?

Please indicate whether the cell lines used in this manuscript utilize telomerase or ALT.

Please discuss whether this method would work in cells that utilize ALT.

Please include in the methods the type of blood collection tubes used.

Reviewer #2 (Remarks to the Author: Overall significance):

In this paper, "High-throughput telomere length measurement at nucleotide resolution using the PacBio high fidelity sequencing platform," Tham, Poon, and colleagues used "telobaits" complementary to the single-stranded G-rich 3' telomeric overhangs, which they used to capture full-length telomeres and then use single-molecule real-time (SMRT) sequencing.

They claim their new method is a valuable tool for population-wide epidemiology studies as well as for the longitudinal assessment of telomere attrition in individuals. They also show extreme heterogeneity of telomeric variant sequences (TVSs) dispersed throughout the telomere repeat region, including a unique distribution pattern and sequence of TVSs in individuals that may be used as a fingerprint for sample identification. Finally, they provide evidence that TVSs disrupt the continuity of the canonical (5'-TTAGGG-3')_n telomere repeats, which affects the binding of shelterin complexes at the chromosomal ends and telomere protection.

Overall, the work is intriguing and opens new directions in research, and the method is well-validated and useful, thus I would only request although some parts of the text need clarification or updating:

- 1) The authors state that previous work has shown that "telomeric repeat regions are interspersed with variant sequences exhibiting extreme heterogeneity, which may arise from errors during the sequencing of such repetitive DNA loci." However, as they and the previous paper showed, this is not a concern with HiFi reads, so I would change this to say "which may be related to sequencing kinetics in repetitive areas," since that is really the question.
- 2) The authors noted that they saw "slight variation in mean observed between independent sequencing runs might be due to varying loading efficiencies, as a loading bias is known to exist for shorter genomic DNA fragments." But, if they normalized each plot by the loading efficiency, could this issue be corrected?
- 3) They found a indicates a negative correlation with $r=-0.447$ and $p\text{-value}=0.001$ for the telomere length and age, but was this better on some chromosomes vs. others?
- 4) When they ran their PBMCs to look at telomeres, were there any differences between males and females?
- 5) Related to #4, were any non-canonical motifs apparently unique to one person, vs. those which were more common?
- 6) Some other literature has shown that telomere length and sequence mapping can also occur on Oxford Nanopore Technologies and show variation in others contexts, and this should be referenced as well:

Reviewer #2 (Remarks to the Author: Impact):

Nature Methods perhaps.

Reviewer #2 (Remarks to the Author: Strength of the claims):

Yes it looks good

Reviewer #2 (Remarks to the Author: Reproducibility):

Need to see the methods and raw data posted - that was not clear.

Responses to Reviewers

Reviewer 1:

1. The abstract claims that “the unique distribution pattern and sequence of TVSs in individuals may be used as a fingerprint for sample identification.” This is a bold claim with wide ranging implications across telomere biology, genetics, and forensics, and has the potential to be picked up in the popular press. However, no data is presented in the manuscript that substantiates this claim. Specifically, although the authors present data that TVSs overall appear to be different across the 48 patient PBMC samples, they do not actually show that if one were to repeat this method with a biological replicate of one of these samples, that it would uniquely match to the same sample.

→ We thank the reviewer for raising this point. To address the reproducibility of our method, we have compared the telomere variant sequence (TVS) signatures (or patterns) of biological replicates from cultured cell lines as well as patient samples sequenced in the same and independent PacBio HiFi sequencing runs. We observed a high concordance of TVS signature amongst biological replicates of cultured cell lines as well as patient samples (Figure S12, S13A-S13B, and Table S6). An example from Chr. 7q is shown here in Figure S12B where cultured cell lines across independent PacBio HiFi sequencing runs shows high correlation scores (>0.9), highlighting their robustness and reproducibility.

Figure S12B: The TVS signature correlation of Chr. 7q amongst biological replicates of cell line samples.

2. Similarly, as this study is limited to only 48 individuals, it is unclear if the authors are merely identifying various telomere haplotypes that are specific to certain populations, or genetic features that are truly unique to each individual. Please adjust this language accordingly, or substantiate this claim by providing reproducible population-level data across biological replicates from diverse cohorts that accounts for haplotype-specific features.

→ We thank the reviewer for raising this important point regarding the uniqueness of telomere haplotypes across individuals and populations. According to our cohort of 104 mixed-ethnicity individuals, the data indicated that TVS signatures in telomere haplotypes are unique to each person. Hence, we are inclined to believe that is the case. Our cohort is primarily made up of Chinese, with 3 Indians, 4 African Americans, 6 Hispanics, and 12 Caucasians (Table S3). As shown in Figure S11, the TVS signatures of telomeres at Chr. 7q are dissimilar amongst individuals, as indicated by modest correlation scores (0.50-0.90). The individual uniqueness of TVS is also observed for telomeres on other chromosomal ends (Table S5). Conversely, the TVS signature comparison amongst biological replicates from different sequencing runs yielded high correlation scores (>0.90) (Figure S13B). We understand that our small cohort represent neither a population-level data nor a diverse cohort, therefore we are not claiming, but rather, suggesting that genetic features in telomeres might be potentially unique to each individual.

Figure S11: The TVS signature correlation for Chr. 7q amongst a mixed ethnicity cohort of patient samples (P01 to P104).

Figure S13B: The TVS signature correlation for Chr. 7q amongst biological replicates from two independent sequencing runs.

3. The abstract claims that “the presence of TVSs disrupts the continuity of the canonical (5'-TTAGGG-3')n telomere repeats, which affects the binding of shelterin complexes at the chromosomal ends and telomere protection.” However, the authors do not actually demonstrate that TVSs disrupt the binding of shelterin complexes at the chromosomal ends or disrupt telomere protection. Rather, the authors show that TVSs disrupt canonical shelterin binding elements. However, the impact of this on shelterin occupancy and telomere protection is not evaluated in this manuscript.

→ We thank the reviewer for raising this question. To address this question, we have employed two independent assays. First, we show that the binding of Shelterin subunits-TRF1 and TRF2 is abolished by TVSs using gel mobility shift assay, as these TVSs failed to compete the binding of TRF1 and TRF2 to ³²P-labeled canonical “TTAGGG” telomere repeats (Figures S14A and S14B). Second, the introduction of such TVSs to the templating region of telomerase RNA (hTR) will result in the incorporation of TVSs onto the tip of chromosome ends by telomerase catalytic subunit¹. Since these TVSs disrupt the binding of shelterin complex in vivo, the telomeres become uncapped, triggering DNA damage responses as indicated by the presence of telomere dysfunctional foci². As shown in (Figures S14C and S14D), telomere dysfunctional foci are readily observed in WA18 (hTR^{-/-}) cells expressing mutant hTR containing TVSs.

4. The calculation of “telomere length” presented in figure 2A is a little awkward and not entirely consistent with standard definitions of telomere length. Specifically, telomere length is often thought of as the continuous linear length of DNA sequence from the start of the TTAGGG repetitive element to the terminal end of a telomere. It is unclear why the authors have chosen to use a non-continuous length measurement. It would be useful for the authors to discuss how their results could differ if they used a continuous linear length definition of telomere length (i.e., “raw telomere length”).

→ We agree with the reviewer that the traditional definition of telomere length ranges from the start of 5'-TTAGGG-3' to the chromosome end. In our study, we arbitrarily defined the 5' start of a telomere to contain at least two consecutive telomeric repeats (5'-TTAGGGTTAGGG-3'). This is due to two reasons:

- 1) Two consecutive telomeric repeats provides the minimum binding motif for the shelterin complex which contains 1.5 canonical repeats 5'-TTAGGGTTA-3'³.
- 2) Using a single 5'-TTAGGG-3' repeat as a start definition may overestimate the length of telomeres as a single 5'-TTAGGG-3' repeat would be found quite frequently in the human genome (about once every 4kb).

Thus, we estimate the telomere length of a read ranging from the first 5' occurrence of at least two consecutive telomeric repeats (5'-TTAGGGTTAGGG-3') to the 3' end of the read just before our capture oligos. We define this length as the **raw telomere length**, which is similar to the traditional definition of telomere length (from start of TTAGGG repeat to the chromosome end).

As we have pointed out in this paper, telomeric region indeed contains TVSSs with extreme heterogeneity. The telomere-associated shelterin complex, which protects the chromosomal ends from being recognized as double-stranded DNA breaks, binds to canonical telomere repeats in a sequence-specific manner that is predicted to be disrupted by the presence of TVSSs (Figure S14)^{4,5}. Therefore, we also calculated the cumulative length of canonical 5'-TTAGGG-3' repeats alone as a form of effective telomere length estimation, which we define as **telomere length**.

We have presented both telomere length (Figure 2, 3, 4, 5, S7 and TableS4) and raw telomere length (Figure S5, S7 and Table S4) estimation and distribution of all cultured cells and patient samples in our paper. Both estimations are comparable to each other.

We propose that the heterogeneous nature of these TVSSs may also contribute to the non-coding polymorphisms that affect the binding density of shelterin complexes to the canonical telomeric (5'-TTAGGG-3')_n repeats and consequently leads to differences in the extent of telomere protection during human aging. Hence, we believe that both the effective telomere length and TVSSs are important factors that may be involved in human aging and diseases.

5. Although this method provides “telomere length measurement at nucleotide resolution”, the authors have not demonstrated that this method can substantially determine which specific telomere a given read arises from. Specifically, the authors only report genomic-location specific results for 15q and 19q (figure 5B), with all other analyses being agnostic of specific genomic location. I suspect that this is at least partially due to the use of hg38 as the reference genome, as telomeric regions within hg38 are quite problematic. However, the lack of genomic position-precise maps beyond 15q and 19q limits the ability to determine the true extent of TVVs within an individual, as it is unclear based on the findings whether these TVVs are fixed within an individual on a given telomere end or are variable across different cells from the same individual. This latter point would have significant implications for using TVVs as a “fingerprinting method”.

→ We thank the reviewer for this comment and we agree that the difficulty of mapping telomeres to their respective chromosomal ends is partially due to problematic telomeric regions in the hg38 reference genome. To improve our mapping efficiency, we turned to using the newly published T2T-CHM13 haploid genome reference⁶. We remapped our telomeres to T2T-CHM13 and was able to align them to >15 specific chromosomal ends confidently for most of the patient samples (Figure below). Our limited mapping capability might be due to short subtelomeric lengths in some of the telomere-containing HiFi reads (~250bp on average, a result of genomic DNA trimming using RsaI and HinfI), which may not be sufficient for reliable mapping to all chromosome ends. However, we are confident that our telomere mapping would be optimized in the near future with several experimental fine-tuning, including the use of a better suited restriction enzyme for genomic DNA trimming.

Having able to map telomeric reads to their respective genomic locations, our subsequent end-specific analyses suggest that TVVs appear to be quite constant across our samples. In our sample cohort, the TVV percentages across the telomeric region of a given telomere end often exhibit 0% (absence of TVVs) and 100% (presence of TVVs), indicating maximum consensus in those regions across all reads (Figures 5B and S10).

We agree with the reviewer that our current telomere end mapping limits the ability to showcase the full extent of TVV across all telomere ends. We hope to resolve this limitation in the near future. For now, we have increased our TVV analysis to additional chromosomal ends such as Chr. 2p, Chr. 3p, Chr. 5p, Chr. 6q, Chr. 7q, Chr11q, Chr. 12q, Chr. 14q, and Chr.17p (Figure S11 and Table S5).

Figure: Telomere mapping and haplotype clustering. Number of telomeric reads mapping to each chromosomal end across 81 PBL samples in Singapore cohort. The subtelomeric region of telomeric reads was mapped to the T2T-CHM13 reference genome.

6. For the reads with only a 12nt telomere length, can the authors address whether the ends of these reads happen to correspond to digestion sites for *RsaI*, *HinfI* or *EcoRI* within TVSSs, as this would indicate a potential artifactual mechanism for these reads? Similarly, do these reads from the same individual have similar sub-telomeric sequences, which would suggest an off target restriction digestion issue as opposed to a shearing issue as is proposed by the authors.

→ We thank the reviewer for raising this point and we agree that there may be an artifactual mechanism for 12nt telomeric reads. After analyzing the ends of 12nt telomeric reads, we found that almost all of them (98%) terminates with the GT motif, which corresponds to the *RsaI* digestion site. Hence, it is likely that these reads might be artificially shortened by *RsaI* digestion within the TVSSs. However, this does not seem to occur to a specific chromosomal end as the subtelomeric sequences of these reads from the same individual were found to be dissimilar. Nonetheless, we acknowledge that, while only affecting a minority of reads, this artificial telomere shortening mechanism is a drawback of our approach, and we are actively investigating other restriction enzymes such as *HindIII*, which seems promising in eliminating this artefact (more details in the response to next comment).

Minor comments:

1. The analysis of G-strand ends does not account for the contribution of potential differences in the ligation efficiencies of the 6 telobaits.

→ We thank the reviewer for raising this reasonable point. To investigate the ligation efficiencies of the 6 telobaits, we analyzed the ends of non-telomere-containing reads, which accounts for more than 60% of the total captured reads. Our results showed that these reads are bias towards having the "TAGGGT" ends (about 50%) (Figure A below), which suggested a ligation bias of the "TAGGGT" telobait. However, we believe that this bias is most likely caused by the *RsaI* restriction enzyme digestion rather than a bias in telobait ligation efficiency. The *RsaI* digestion would generate a higher proportion of DNA fragments ending with "GT", and together with a chance of DNA end unzipping, these reads would be captured in higher amounts than others.

To validate our hypothesis, we swapped out *RsaI* for *HindIII* in the enrichment of six of our patient samples and found that the "TAGGGT" end bias is gone (Figure B below). Consistent with *RsaI* digested samples, the telomeric G-strand ends of *HindIII* digested samples also show a preference for 5'-GGTTAG-3' (60%) (Figure C below), which is to be expected as 5'-GGTTAG-3' is predominantly generated during telomerase-dependent telomere elongation.

Hence, these results show that the preferential 5'-GGTTAG-3' G-strand end observed in our data is not due to a bias in ligation efficiency of telobaits.

Figure: The telomeric G-rich strand end preferentially with 5'-GGTTAG-3'. **A.** The ligation efficiency of six telobaits compiled from non-telomere-containing PacBio HiFi reads digested with *RsaI* during the genomic DNA digestion and telobait ligation step. **B.** The ligation efficiency of six telobaits compiled from non-telomere-containing PacBio HiFi reads digested with *HindIII* during the genomic DNA digestion and telobait ligation step. **C.** Bar graph showing the percentage of different G strand end sequences across patient PBL samples when *HindIII* was used in the genomic DNA digestion and telobait ligation step. The most common G strand end observed is 5'-GGTTAG-3'.

2. If available, the ethnicity of the 48 individuals should be reported to aid in future studies integrating this data.

→ We thank the reviewer for the suggestion. The ethnicity of all patient samples is shown in Table S3.

3. Please add additional sequencing details such as:

3.1. Number of ZMWs with sequence information from the 48-sample multiplexed run (based on figure 3A, it appears that only ~250,000 reads were recovered, was the SMRT cell underloaded, or were there a lot of reads that failed to get CCS coverage?);

→ The number of ZMWs with sequence information for all four PacBio HiFi runs used in this paper is shown in Table S1. The low recovery of reads is likely attributed to both flowcell underloading and reads failing to get CCS coverage.

3.2. Per-read CCS coverage versus CCS read length plot from the aforementioned run;

→ Per-read CCS coverage versus CCS read length plot of all four PacBio HiFi runs used in this paper is shown in Figure S20.

3.3. Demultiplexing efficiency from the aforementioned run (i.e., how many reads were demultiplexed versus those that were unable to be demultiplexed). For the sake of reproducibility, please elaborate on these details in the Methods for further consideration at Communications Biology and Nature Communications. Please also review the Open Research Evaluation at the end of this document for other suggestions to improve transparency and data reporting.

→ The demultiplexing efficiency of all four PacBio HiFi runs used in this paper is shown in Table S1. The demultiplexing process was performed using our in-house bioinformatics tool, Telomap, which is freely available at <https://github.com/cytham/telomap>. We have elaborated Telomap's workflow in the Methods section under "Analysis of Telobait-enriched PacBio HiFi sequencing data".

3.4. Figure S1C, what is the etiology of those three spikes in the histogram on the right?

→ We thank the reviewer for this observation. The three spikes of error frequency coincide with three non-canonical telomeric repeats (5'-TTTAGAG-3') highlighted in green below. As a result, the error spikes are most likely caused by artefactual misalignments by the sequence aligner tool (Pbmm2), where the read alignment might be shifted by one or more telomeric repeat motifs, resulting in a forced pairing between a canonical and a non-canonical repeat, producing a high mismatch error rate of A to G in these three regions. On the contrary, the three non-repetitive regions (unhighlighted) coincide with the three error frequency dips, indicating low error in these unique regions.

>spike-in_pWY82

```
TTTATTTAGGGTTTAGGGTTTAGGGTTTAGGGTTTAGGGTTTAGGGTTTAGGGTTTAGGGTTTAGGGTTTAGGGTTA
GGGTTTAGGGTTTAGGGTTTAGGGTTTAGGGTTTAGGGTTTAGGGTTTAGAGTTTAGGGTTTAGGGTTTAGGGTTTAG
GGTTTAGGGTTTAGGGTTTAGGGTTTAGGGTTTAGGGTTTAGGGTTTAGGGTTTAGGGTTTAGGGTTTAGGGTTAG
GGTTTAGGGTTTAGGGTTTAGGGTTTAGGGTTTAGGGTTTAGGGTTTAGGGTTTAGGGTTTAGGGTTTAGGGTTAG
GGTTTAGGGTTTAGGGTTTAGGGTTTAGGGTTTAGGGTTAGGGTGACCTGCAGCCCAAGCTCTCGGGTCCCACAGC
GCTTATTTAGGGTTTAGGGTTTAGGGTTTAGGGTTTAGGGTTTAGGGTTTAGGGTTTAGGGTTTAGGGTTTAGGGTT
TAGGGTTTAGGGTTTAGGGTTTAGGGTTTAGGGTTTAGGGTTTAGGGTTTAGGGTTTAGAGTTTAGGGTTTAGGGTT
AGGGTTTAGGGTTTAGGGTTTAGGGTTTAGGGTTTAGGGTTTAGGGTTTAGGGTTTAGGGTTTAGGGTTTAGGGTTA
GGGTTTAGGGTTTAGGGTTTAGGGTTTAGGGTTTAGGGTTTAGGGTTTAGGGTTTAGGGTTTAGGGTTTAGGGTTA
GGGTTTAGGGTTTAGGGTTTAGGGTTTAGGGTTTAGGGTTTAGGGTTTAGGGTTTAGGGTTTAGGGTTAGGGTGAC
CTGCAGCCCAAGCTCTCGGGTCCCACAGCGCTTATTTAGGGTTTAGGGTTTAGGGTTTAGGGTTTAGGGTTTAGGGT
TTAGGGTTTAGGGTTTAGGGTTTAGGGTTTAGGGTTTAGGGTTTAGGGTTTAGGGTTTAGGGTTTAGGGTTTAGGGT
TAGAGTTTAGGGTTTAGGGTTTAGGGTTTAGGGTTTAGGGTTTAGGGTTTAGGGTTTAGGGTTTAGGGTTTAGGGTT
AGGGTTTAGGGTTTAGGGTTTAGGGTTTAGGGTTTAGGGTTAGGGTGACCTGCAGCCCAAGCTCTCGGGAAATCACTAGTGAT
CGATATCTCCCTTAGTGAGGGTTAATGTGAATTT
```

Legend:

Yellow – Canonical *Arabidopsis* telomeric repeat (TTTAGGG)

Green – Non-canonical telomeric repeat (TTTAGAG)

Unhighlighted – Non-telomeric sequence

3.5. Please indicate whether the cell lines used in this manuscript utilize telomerase or ALT. Please discuss whether this method would work in cells that utilize ALT.

→ We thank the reviewer for this question. The cell lines used for telomere length measurement in this manuscript are either telomerase-positive cancer cell lines (HCT116, 293T and T24) or telomerase negative primary cells (WI-38 and IMR90).

PacBio HiFi sequencing reads are not suitable for telomere length measurement in ALT cells at this moment. ALT cells have very heterogeneous telomere length distribution with extremely long telomeres (Figure 1B) that is beyond the optimal processivity of PacBio HiFi DNA polymerase. We have pointed out this as a potential limitation of PacBio HiFi platform in the discussion.

3.6. Please include in the methods the type of blood collection tubes used.

→ We thank the reviewer for this question. The details of the type of blood collection tubes and genomic DNA purification protocol is now included in the Methods.

Review 2:

1. The authors state that previous work has shown that “telomeric repeat regions are interspersed with variant sequences exhibiting extreme heterogeneity, which may arise from errors during the sequencing of such repetitive DNA loci.” However, as they and the previous paper showed, this is not a concern with HiFi reads, so I would change this to say “which may be related to sequencing kinetics in repetitive areas,” since that is really the question.

→ We thank the reviewer for this suggestion. We have changed the wording to “the telomeric repeat regions are interspersed with variant sequences exhibiting extreme heterogeneity, which may be related to sequencing kinetics in repetitive DNA loci.”

2. The authors noted that they saw “slight variation in mean observed between independent sequencing runs might be due to varying loading efficiencies, as a loading bias is known to exist for shorter genomic DNA fragments.” But, if they normalized each plot by the loading efficiency, could this issue be corrected?

→ We thank the reviewer for this good suggestion and believe that the slight mean variation might be corrected by loading efficiency normalization. However, we are not aware of any published method to measure and correct the loading efficiency of each sequencing run, and hence regret that we are not able to incorporate the reviewer’s suggestion into our analysis. Nonetheless, our findings suggest that such bias does not seem to limit the reproducibility of accurate telomere length measurement when compensated by increasing read depth of samples.

3. They found a indicates a negative correlation with $r=-0.447$ and $p\text{-value}=0.001$ for the telomere length and age, but was this better on some chromosomes vs. others?

→ We thank the reviewer for raising this question. While we are able to map the telomere-containing PacBio HiFi reads to a subset of chromosomal ends based on the latest human genome assembly (T2T-CHM13)⁶, the limited number of reads from each chromosomal ends in each sample prevent accurate telomere length estimation. We will further address this question in our follow up studies.

4. When they ran their PBMCs to look at telomeres, were there any differences between males and females?

→ We thank the reviewer for raising this question. Indeed, we did see a trend in our samples indicating that females have longer telomere than males (Figure below). However, it is not statistically significant due to the low number of samples that we have analyzed.

Figure: The mean telomere length of male and female from the 81 patient PBL samples derived in Singapore cohort.

5. Related to #4, were any non-canonical motifs apparently unique to one person, vs. those which were more common?

→ We thank the reviewer for this interesting question. We reanalyzed our data and did not find any 6 nt non-canonical motifs that is overly represented (more than 5%) in any individual. Across our samples, we observed that the proportions of the common 6 nt non-canonical motifs per sample fairly resembles the collective proportions shown in Figure 4B. Interestingly, we found that several samples have a slightly different distribution profile of common non-canonical motifs. For instance, samples P04, P26, P27, P28, P29, P42, P47, P86, P87 and P88 possess higher amounts of “TTCGGG” motifs than others (Figure below).

Figure: Non-canonical telomeric motif distribution across 104 patient samples. The common non-canonical motifs were queried for their proportions in each sample. None of the samples possess other non-canonical motifs that is greater than 5% in proportion.

6. *Some other literature has shown that telomere length and sequence mapping can also occur on Oxford Nanopore Technologies and show variation in others contexts, and this should be referenced as well: [https://www.cell.com/cell-reports/fulltext/S2211-1247\(20\)31446-7](https://www.cell.com/cell-reports/fulltext/S2211-1247(20)31446-7) and [https://www.cell.com/cell-reports/fulltext/S2211-1247\(20\)31424-8](https://www.cell.com/cell-reports/fulltext/S2211-1247(20)31424-8) For the sake of context, please incorporate these references into the Discussion or Introduction.*

→ We thank the reviewer for raising this suggestion. We have incorporated these two references in our revised manuscript.

7. *Need to see the methods and raw data posted - that was not clear. This point was also raised by Reviewer #1.*

→ We thank the reviewer for raising this point. We have amended our methods section in the manuscript to provide detailed experimental procedure. The raw data are also provided in Supplementary Figures S17-S19.

8. *Please clarify in the Data Availability statement where source data underlying Fig 2c, 2e, 3c-d, 4, and 5c can be obtained.*

→ We have clarified in the Data Availability statement that the source data for these Figure panels can be found in Table S5.

→ The source data for Figures 2C, 2E, 3A, 3B, 3C, 4A, 4B, 5D, S7, S11 S12B and S13B can be found in Table S5.

Please include the full, uncropped blot/gel images for Fig 1b and 2f as Supplementary Figures (complete with titles and legends) and cite the new Supplementary Figures in the main manuscript text.

→ The raw data are provided in Supplementary Figures S17-S19.

Cell line misidentification and cross-contamination is a common problem with serious consequences. Authors are asked to report on the source and authentication of their cell lines.

→ The cell lines presented in this manuscript: HCT116, 293T, T24, IMR90 and WI38 were purchased from ATCC; WA18 embryonic stem cell line is purchased from WiCell.

Please expand on the Southern blot hybridization protocol

→ We have expanded the Southern blot hybridization protocol as requested in the Methods section.

References:

1. Li S, *et al.* Rapid inhibition of cancer cell growth induced by lentiviral delivery and expression of mutant-template telomerase RNA and anti-telomerase short-interfering RNA. *Cancer Res* **64**, 4833-4840. (2004).
2. Takai H, Smogorzewska A, de Lange T. DNA damage foci at dysfunctional telomeres. *Curr Biol* **13**, 1549-1556 (2003).
3. Hanaoka S, Nagadoi A, Nishimura Y. Comparison between TRF2 and TRF1 of their telomeric DNA-bound structures and DNA-binding activities. *Protein Sci* **14**, 119-130 (2005).
4. Broccoli D, Smogorzewska A, Chong L, de Lange T. Human telomeres contain two distinct Myb-related proteins, TRF1 and TRF2. *Nat Genet* **17**, 231-235 (1997).
5. Konig P, Fairall L, Rhodes D. Sequence-specific DNA recognition by the myb-like domain of the human telomere binding protein TRF1: a model for the protein-DNA complex. *Nucleic Acids Res* **26**, 1731-1740

(1998).

6. Nurk S, *et al.* The complete sequence of a human genome. *Science* **376**, 44-53 (2022).

Reviewer comments:

Reviewer #1 (Remarks to the Author: Overall significance):

I greatly appreciate the amount of effort and work that the authors put into this revision. They have significantly strengthened the manuscript and addressed many of my initial concerns regarding their conclusions.

I have a couple of remaining points below that should be addressed:

Regarding figures S11, S12B, and S13B, how are the two haplotypes from each diploid individual handled? It appears that they are not being considered. As the two haplotypes from each individual should be as divergent as two haplotypes from separate individuals, this could cause issues in terms of this analysis.

Addressing the question regarding population stratification of TVVs for Figure S11 would be better performed using a dendrogram rather than a heatmap, with individuals of different genetic ancestry separately colored to determine whether there is population structure to TVV sequence.

In regards to the 12nt telomere repeats, the manuscript text still states "This is likely due to the criterion we used to determine the 5' start of a telomere or could be the result of random DNA shearing during purification." Please update this section of the text in light of the finding that 98% of these 12nt telomeres result from an artifact of RsaI digestion.

Reviewer #3 (Remarks to the Author: Overall significance):

I was asked to review the rebuttal of reviewer 2. I am not providing any comments of the overall paper. Here are my comments on the rebuttal for reviewer 2 comments:

1, 6, 7, 8 and the last few comments are minor changes and the authors addressed them properly.

Comment 2, a question that I find is not a critical one because loading variation is common for most sequencing platform, including PacBio and does not impact reproducibility of the data. I agree with authors response.

Comment 3, this is a great question and the author was not able to provide satisfactory answer. It seems it will require major reanalysis that may be outside of the scope of this study.

Comments 4, required to conduct a significance test and the author have done it correctly to address the reviewer's point.

Comment 6, the reviewer asked to find 6nt non-canonical motif enrichment and the authors did show with reanalysis, some enrichment that are not significant. This point was addressed correctly.

Based on reviewer's 2 comment, the authors made good attempt to revise the manuscript and it definitely made it better.

Reviewer #4 (Remarks to the Author: Overall significance):

They can specifically and efficiently capture full-length telomeres and subject them to single-molecule real-

time (SMRT)

sequencing, which provides the necessary accuracy and throughput to uncover the steady state telomere length distribution in human culture cell lines as well as clinical patient samples. This provides an alternative approach to quantify the number of chromosomes.

Reviewer #4 (Remarks to the Author: Impact):

There are significant challenges in quantifying telomere length. The paper reports the use of high-throughput telomere length measurement at nucleotide resolution using PacBio HiFi reads. They found extreme heterogeneity of telomeric variant sequences (TVSs) that is dispersed throughout the telomere repeat region.

I focused on the response to reviewers.

Reviewer 2:

Regarding the cell lines and origins, the authors provided response. However, were the passage information provided? Additionally, the reviewer was asking “the source and authentication” of the cell lines, not just where they are purchased. Where any authentication done to ensure that these cell lines are indeed that ones that are expected without sample mix-up?

Reviewer #4 (Remarks to the Author: Strength of the claims):

Overclaimed in many instances.

Reviewer #4 (Remarks to the Author: Reproducibility):

I do not have concern on this.

Responses to Reviewers

Reviewer 1:

1. Regarding figures S11, S12B, and S13B, how are the two haplotypes from each diploid individual handled? It appears that they are not being considered. As the two haplotypes from each individual should be as divergent as two haplotypes from separate individuals, this could cause issues in terms of this analysis.

→ We thank the reviewer for raising this point. The two haplotypes in each diploid individual of a particular chromosome end were identified by K-means clustering (k=2) and subsequently balanced in read counts before combining them to produce a single TVS signature plot. For example, 40 reads are detected at Chr. 1p of a sample. Out of the 40 reads, two clusters of sizes 22 and 18 reads were identified, which represents the two haplotypes. As a result, the final reads utilized for analysis were composed of 18 reads from Cluster 1 and all 18 reads from Cluster 2, for a total of 36 reads used to generate the TVS signature plot. We agree that the two haplotypes of an individual are divergent, and thus, combining the two haplotypes into a single TVS signature plot would further increase its uniqueness for each sample. We hope that this clearly explains how we handle the haplotypes in our analyses.

2. Addressing the question regarding population stratification of TVSs for Figure S11 would be better performed using a dendrogram rather than a heatmap, with individuals of different genetic ancestry separately colored to determine whether there is population structure to TVS sequence.

→ We thank the reviewer for this wonderful suggestion. We have performed hierarchical clustering on the heatmap and generated the dendrograms for TVS signature of each chromosomal end. The dendrogram for the TVS signature of Chr. 7q is now presented in Figure S11 with ethnicity colored (shown below). The dendrograms for the remaining mapped chromosomal ends are inserted in the “Table S5 TVS signature”. However, we did not observe any obvious ethnicity pattern for TVS signature.

3. In regards to the 12nt telomere repeats, the manuscript text still states "This is likely due to the criterion we used to determine the 5' start of a telomere or could be the result of random DNA shearing during purification." Please

update this section of the text in light of the finding that 98% of these 12nt telomeres result from an artifact of RsaI digestion.

→ We thank the reviewer for raising this point. We have changed the statement for the origin of the 12nt telomere repeats to: "This is likely due to the artifact from RsaI digestion as 98% of the ends of 12nt telomeric reads terminate with "GT", which corresponds to the RsaI digestion site." as suggested by the reviewer 1, in page 11 of main text.

Reviewer 4:

Regarding the cell lines and origins, the authors provided response. However, were the passage information provided? Additionally, the reviewer was asking "the source and authentication" of the cell lines, not just where they are purchased. Where any authentication done to ensure that these cell lines are indeed that ones that are expected without sample mix-up?

→ We thank the reviewer for raising this point. The cell lines used for our studies are all in early passages (<5 passages) since their purchase from ATCC except for WI-38 (population doubling 40) and IMR90 (population doubling 47), which have been passaged extensively *in vitro*.

To further authenticate the identifies of the cell lines used in our studies (including 293T, HeLa, T24, HCT116, IMR90, WI38, VA13 and U2OS), we have sent the genomic DNA from these cell lines to 1st BASE for short tandem repeat (STR) profiling. Twenty-four STR loci plus gender determining locus, Amelogenin, were amplified using commercially available GenePrint 24 system from Promega. The PCR products were processed using Applied Biosystems™ DNA analyzer. Data were analyzed using GeneMapper v4.0 software (Applied Biosystems™) and compared with the existing STR database of ATCC human cell lines. We are glad to show that all the cell lines have matched the specific STR profile of the indicated cell lines as shown in the "Cell line authentication report".

REVIEWERS' COMMENTS:

Reviewer #1 (Remarks to the Author: Overall significance):

I am satisfied with the updated responses and analyses.